# On the nexus between landslide susceptibility and transport infrastructure – an agent-based approach

Matthias Schlögl[1,2,3,4], Gerald Richter[5], Michael Avian[4], Thomas Thaler[3], Gerhard Heiss[6], Gernot Lenz[5], and Sven Fuchs[3]

[1]Transportation Infrastructure Technologies, Austrian Institute of Technology (AIT), Vienna, Austria
[2]Institute of Applied Statistics and Computing, University of Natural Resources and Life Sciences (BOKU), Vienna, Austria
[3]Institute of Mountain Risk Engineering, University of Natural Resources and Life Sciences (BOKU), Vienna, Austria
[4]Staff Unit Earth Observation, Zentralanstalt für Meteorologie und Geodynamik (ZAMG), Vienna, Austria
[5]Dynamic Transportation Systems, Austrian Institute of Technology (AIT), Vienna, Austria
[6]Environmental Impact Assessment, Austrian Institute of Technology (AIT), Vienna, Austria

*Correspondence to:* Matthias Schlögl (matthias.schloegl@boku.ac.at)

**Abstract.**

Road networks are complex interconnected systems. Any sudden disruption can result in debilitating impacts on human life or the economy. In particular, road systems in mountain areas are highly vulnerable, because they often do not feature redundant elements at comparable efficiency.

This paper addresses the impacts of network interruptions caused by landslide events on the (rural) road network system in Vorarlberg, Austria.

Based on a landslide susceptibility map we demonstrate the performance of agent-based traffic modeling using disaggregated agent data. This allows gaining comprehensive insights into the impacts of road network interruptions on the mobility behavior of affected people. Choosing an agent-based activity-chain model enables us to integrate the individual behavioral decision-
making processes in the traffic flow model. The detailed representation of individual agents in the transport model allows for optimizing certain characteristics of agents and to include their social learning effects into the system.

Depending on the location of the interruption, our findings reveal median deviation times ranging between several minutes and more than half an hour, with effects being more severe for employed people than for unemployed individuals.

Moreover, results show the benefits of using agent-based traffic modeling for assessing the impacts of road network inter-
ruptions on rural communities by providing insight into the characteristics of the population affected, as well as the effects on daily routines in terms of detour costs. This allows hazard managers and policy makers to increase the resilience of rural road network systems in remote areas.

## 1 Introduction

Infrastructure networks and related assets support the delivery of essential goods and services to society (European Commis-
sion, 2017; Mejuto, 2017; Gutiérrez and Urbano, 1996). In particular, the functionality of socio-economic systems in modern communities heavily depends on extensive, interconnected transport networks because any disruption may cause rippling ef-

fects, eventually entailing instability of other critical infrastructure – both domestically and beyond (Bíl et al., 2015; Jaiswal et al., 2010). Main challenges are negative socio-economic consequences (high direct and indirect losses) to societies as a result of hazard events (Bordoni et al., 2018; Rheinberger et al., 2017; Pfurtscheller and Vetter, 2015; Kellermann et al., 2015; Pachauri and Meyer, 2014; Schweikert et al., 2014; Pfurtscheller, 2014; Meyer et al., 2013; Pfurtscheller and Thieken, 2013;

Nemry and Demirel, 2012; Taylor and Susilawati, 2012; Rheinberger, 2011; Jenelius, 2009; Koetse and Rietveld, 2009).

The impacts caused by severe weather events and associated hazards underline the importance of resilient and reliable transportation infrastructure (Eidsvig et al., 2017), especially in complex landscapes such as the European Alps where the topography impedes redundancies and alternative routing. Failure and disruption of transport infrastructure can therefore affect a broader environment due to cascading effects which result from the dependence of economies, institutions and societies on

such networks (Kellermann et al., 2015; Doll et al., 2014; Keller and Atzl, 2014; Pfurtscheller, 2014; Meyer et al., 2013; Kappes et al., 2012). This is especially true under severe weather conditions triggering disasters, because reliable networks are crucial for emergency response to avert further damage, save lives and mitigate economic losses. Network reliability in this context is defined to comprise network availability and network safety. Non-reliable transportation networks and the associated overall societal loss introduced by destructive incidents considerably exceeds the mere physical damage to such infrastructure. Apart

from an impairment of roads – which results in maintenance and reconstruction efforts to be carried out by road operators (c.f. Donnini et al., 2017) – secondary effects such as intangible and indirect costs of damage to infrastructure networks have to be considered in a broader economic context and lead to considerable vulnerability of societies affected (Klose et al., 2015; Pfurtscheller and Thieken, 2013; Meyer et al., 2013; Fuchs et al., 2011; Fuchs, 2009). Consequently the assessment of transport network systems has gained relevance in academia as well as the policy agenda of authorities across all scales (Pant et al., 2018;

Unterrader et al., 2018; Bíl et al., 2017; Pregnolato et al., 2017; Winter et al., 2016; Rupi et al., 2015; Jenelius, 2009; Taylor et al., 2006; Zischg et al., 2005a, b; D'Este and Taylor, 2003; Berdica, 2002).

Since no context-free definition of road network vulnerability exists, respective methodological approaches (even if highly sophisticated) remain fragmentary and repeatedly tailored to individual settings (Bagloee et al., 2017; Eidsvig et al., 2017; Mattsson and Jenelius, 2015; Rupi et al., 2015; Fuchs et al., 2013).

Berdica (2002, p.119), for example, suggested that network vulnerability should be understood as 'susceptibility to incidents that can result in considerable reductions in road network serviceability'. This includes a focus of assessment on the most critical hotspots (links or nodes) within a current network system, where the highest socio-economic impact can be observed, which – according to other scholars – equals exposure (Unterrader et al., 2018; Khademi et al., 2015; Jenelius et al., 2006). On the other hand, Taylor et al. (2006) understood network vulnerability as a close concept to network weakness and thus as the

consequence of failure to provide sufficient capacity for the 'original' purpose of the system, being, to transfer people and goods from point A to point B. This already shows the close connection of network vulnerability to other terms, such as accessibility, remoteness or robustness, which is linked to the idea of network performance (Yin et al., 2016; Taylor et al., 2006; D'Este and Taylor, 2003). In sum, the idea behind vulnerability is a decline in the 'original' capacity to handle the network flow based on disruption (Yin and Xu, 2010). Nevertheless, in the literature two main directions within network vulnerability assessment

can be distinguished: (1) topological vulnerability analysis, which includes the assessment of real transport network systems

(represented in an abstract network); and (2) system-based vulnerability analysis, which focuses on the structure of the network within demand/supply models (Mattsson and Jenelius, 2015). In the context of the present paper, we understand vulnerability as the assessment of the disruptive impact based on a certain event (incident) which causes a malfunction or breakdown in the current road network system (Postance et al., 2017; Pregnolato et al., 2017; Klose et al., 2015; Mattsson and Jenelius, 2015). The potential disruption may span from natural hazard events to terrorist attacks, infrastructure collapses or ordinary traffic accidents (Bagloee et al., 2017; Unterrader et al., 2018; Vera Valero et al., 2016; Mattsson and Jenelius, 2015; Koetse and Rietveld, 2009; Zischg et al., 2005b; Margreth et al., 2003). Depending on the threat, the potential consequence can result in additional travel time from some minutes to total cut-offs of a community (Rupi et al., 2015; Taylor and Susilawati, 2012; Jenelius, 2009; Zischg et al., 2005b). Therefore, a central goal of vulnerability assessment is the identification of the critical links within the current network system that are highly susceptible to such disruptions (Gauthier et al., 2018; Jenelius et al., 2006; Berdica, 2002). In contrast to the ongoing vulnerability debates in natural hazard and risk management of buildings, (see for example Papathoma-Köhle et al. (2017); Fuchs et al. (2011) or Fuchs (2009)) however, network vulnerability usually does not account for any probability of disruption within the assessment (Rupi et al., 2015).

Two main methodological approaches on how to assess road vulnerability exist (Mattsson and Jenelius, 2015; Hackl et al., 2018). The first is a topological one which focuses on characteristics of the road network's links. It is based on graph theory which is widely used in various disciplines, such as computer science, physics, sociology and transportation (Heckmann et al., 2015; Phillips et al., 2015), with the aim to assess and understand networks and their individual properties (Slingerland, 1981). Using graph theory in vulnerability assessment of road networks generally means focussing on specific graph edges (links) and nodes, their criticality or redundancy, to reflect resilience and interdependencies between parts of the network, as well as potential cascading effects (Pant et al., 2016; Rupi et al., 2015; Tacnet et al., 2013; Jenelius et al., 2006; Meyer et al., 2013). This approach, however, is limited by the reduction to connectivity within a network, therefore not including the behavioural aspects of transport network users.

A second group of models bridges this gap by considering link properties and traffic demands on the links of traffic networks. The network loads, together with appropriate traffic dynamics result in alterations of network properties, which gives rise to various stress response effects that can also be observed in real-world traffic. These models differ with respect to the chosen granularity, and can be divided in macro-, meso- and microscopic models (Treiber and Kesting, 2013; Hoogendoorn and Bovy, 2001).

*Macroscopic traffic models* stem from the concept of flow theory, and consider aggregate continuous flow densities of anonymous users on the network. They can be applied to find equilibrium loading states within these networks, as well as to describe dynamic effects within the flow continuum. Their application usually requires solving systems of coupled equations. In contrast, the fine-grained *microscopic traffic models* consider each transportation network user as an individual entity ('agent' i.e. vehicle or pedestrian) with separate interaction details and decisions. These models are implemented as simulation frameworks, iterating over time steps the entire network evolution. Thus, individual entities (agents) retain their specific characteristics throughout the traversal of the network and therefore can react to different circumstances based on these characteristics. *Mesoscopic traffic models* are hybrids between macro- and microscopic models. They are less fine grained and

borrow some characteristics from both approaches, offering a description that is less detailed in time or space, but also less demanding regarding the computational requirements. Depending on the implementation, micro- and mesoscopic models can be 'agent-based', thus retaining the individuality of their agents throughout the model evolution. A more detailed conceptual distinction of agent based models regards the scheduling of mobility demands. Simpler approaches define individual (or multi-

ple unrelated) trips between origin and destination pairs ('trip-based'), whereas more recent frameworks allow the expression of agent activity plans or chains ('activity-based') to be fulfilled by adaptively traversing the transportation networks of the simulation.

The change between levels of granularity in the description of model entities is referred to as (dis-)aggregation for in- or decreasing detail, respectively.

With the modeling discrimination provided above, the approaches of the second group of road vulnerability assessment methods allow to explore the effects of landslide events on a given population and its sub-groups with respect to their mobility requirements. The main drawbacks of aggregated (macroscopic) traffic models in that context include: (1) loss of population individuality, (2) therefore lack of behavioral alterations and co-dependent learning effects of individuals, (3) more time-averaging aspects, prohibiting re-decisions based on incidents, (4) more space-averaging aspects, prohibiting investigation of

localized events without re-building the overall model (e.g. new zoning structure), (5) connected to unavailable consequences of precise socio-demographic measures, (6) macroscopic models are trip-based, considering individual journeys instead of whole day-plans and (7) macroscopic models are adaptable to the increasing level of detail available through continuously improving data by layering of multiple models. Choosing an agent-based activity chain model, which integrates the dynamic aspects of each agent, can overcome these limitations. The vulnerability assessment utilizing activity chain traffic modeling allows

simulations to integrate multiple phenomena, to understand the dynamic interactions of human behavior and the environment in the sense of consequences for households or wider socio-economic systems.

The focus of this paper is on landslide hazards, which repeatedly jeopardized the integrity of road infrastructure by causing structural damage and interruptions (Postance et al., 2017; Klose et al., 2015; Bíl et al., 2014). In the Austrian Alps, 1 444 damaging events to rural roads were recorded in the provinces of Salzburg (2007–2010) and Styria (2008–2011), and debris

flows and landslides caused nearly 50 % of the recorded damage costs (König et al., 2014b). The prevailing hazard potential caused by landslides is aggravated by findings of several other recent studies which have shown that landslide activity and thus related damage will most probably increase with progressing climate change (Schlögl and Matulla, 2018; Gariano and Guzzetti, 2016; Bíl et al., 2015; Klose et al., 2015; Strauch et al., 2015; König et al., 2014a; Keiler et al., 2010). Similar results are available from other mountain regions (e.g. Postance et al., 2017; Unterrader et al., 2018; Meyer et al., 2015; Fuchs et al.,

30 2013).

So far, most studies have mainly focused on primary road networks (Postance et al., 2017; Taylor et al., 2006) and urban areas (Gauthier et al., 2018), while federal and local road networks have been largely neglected. Mountain roads, in contrast to lowland roads, are highly vulnerable due a higher probability of climate-driven hazard events and the inherent obstacles of implementing redundant systems (Schlögl and Matulla, 2018; Matulla et al., 2017; Schlögl and Laaha, 2017; Doll et al., 2014;

Eisenack et al., 2011). Consequently, misleadingly termed as 'forgotten road system', local road networks in fact connect rural

communities in various ways – from supply reliability over public health and tourism to all sorts of economy. Furthermore, mostly issues on technical realization of mitigation and road maintenance have been addressed, rather than socio-demographic impacts on communities or exposed societies (Mattsson and Jenelius, 2015). This paper partly contributes to close the gap by including the full road network system. In particular, the relation between infrastructure and communal development in mountain areas is not one-directional, meaning that it is only the former that can impact the latter; instead, the influence is rather two-way (Jaafari et al., 2015).

The presented approach is complementary to previous studies because of the consideration of whole-day travel plans (as opposed to a focus on peak traffic flow periods on the investigated network), with these plans stemming from the underlying agents' activity chain model. This schedule of activities, which is far less dependent on fixed locations, allows for a more inclusive and flexible reassignment of mobility needs and resulting traffic demands. Therefore, integrating transport route finding and satisfaction of individual activity needs in one single simulation framework facilitates a more detailed and realistic representation of traffic loads on the network. We demonstrate an appropriate methodological response to foreseeable demands imposed by the increasing detail of available mobility data, which brings about particular relevance of this approach for future applications.

The applicability of the approach is demonstrated by the example of Vorarlberg, the westernmost province of Austria (Figure 1). While being the second-smallest federal state, the population density of Vorarlberg is only surpassed by Austria's capital, Vienna, which indicates the need for a resilient transport network. The main traffic artery in this almost completely mountainous area is the connection from Germany to Western Austria, via Rhine valley, Walgau, Klostertal and the Arlberg massif. Apart from this link, which is realized as motorway (A14 and S16), rural roads prevail in the complexly structured topography of Vorarlberg. Because Vorarlberg is almost entirely surrounded by mountain areas and a considerable exposure to extremely high rainfall (with average annual precipitation totals exceeding 2000 mm), the transport system of Vorarlberg is highly exposed to landslides. The combination of (i) being characterized by high landslide susceptibility, (ii) exhibiting a high population density and (iii) lacking alternative routes on the rural network due to the mountain orography makes Vorarlberg a perfect case study.

## 2   Data and methods

Methodologically the approach presented in this paper is divided into two modeling sections:

1. compilation of a landslide susceptibility map in order to identify potential blockage sections in the rural road network, and

2. implementation of agent-based transport simulations to derive impacts of interruptions on local communities. By modeling the responses of individuals to network disturbances the transport model allows for optimizing certain characteristics of agents (e.g. time of departure, route choice, activity list, etc.). Generalized costs of interruptions (i.e. monetary costs, time losses, etc.) are obtainable by employing a utility function to the agents' resulting behavior.

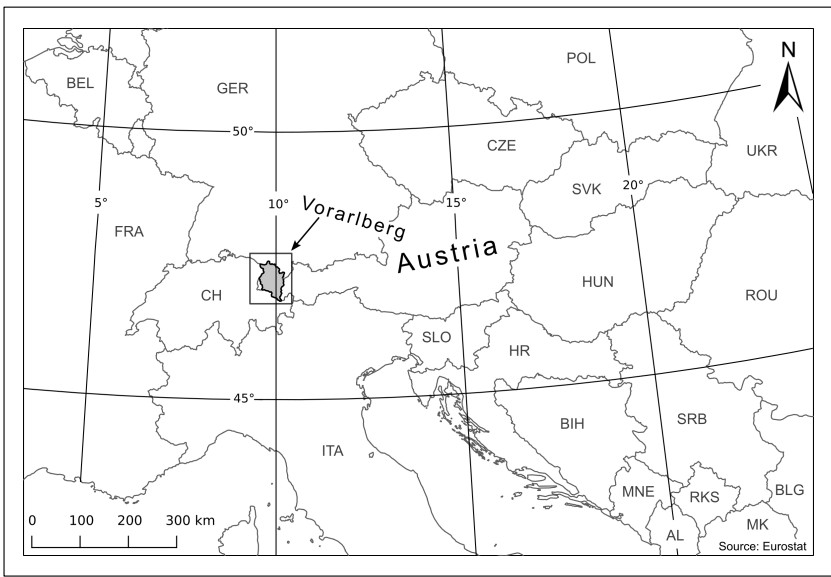

**Figure 1.** Location of the study area in Europe.

## 2.1 Modeling landslide susceptibility

The most decisive data needed for a statistical modeling of landslide susceptibility is an accurate and representative landslide inventory (Zêzere et al., 2017). Different types of landslide inventories exist: (i) Historical data from archives, (ii) field mapping results, (iii) information derived from remote sensing data, and (iv) combined inventories (Bell et al., 2012). In general,

inventories which are based on archive data do only include reported events. These often caused some kind of loss were therefore recorded; landslides occurring in remote areas without significant loss are usually not documented. Event inventories give detailed information about the process location, the process type, the trigger (e.g. heavy rainfall), and sometimes additional data such as a precise event date. In terms of data acquisition inventories based on remote sensing data can be distinguished in two groups: Inventories can either be derived from passive optical sensors (i.e. ortho-images and digital terrain models

(DTMs) obtained through photogrammetry) or from active sensors (e.g. laserscanning, radar). Data from passive systems, such as ortho-images, imply the considerable disadvantage of only sensing the surface from one point per image compared to e.g. laserscanning (constant sensing). Therefore existing vegetation cover often prohibits an area-wide, spatially explicit mapping of landslides, even in stereoscopic analysis (Petschko et al., 2015). In the last decade the availability of DTMs derived from airborne laserscanning (ALS) offers a new quality of high resolution terrain representation (spatial resolution of 1 m). Penetra-

tion of laser pulses results in precise and accurate 3D-information of the surface – even in areas with high vegetation density – and have thus been used in many studies (Proske and Bauer, 2016; Petschko et al., 2015, 2013a). The most promising approach for the generation of representative event inventories in forested and unforested landscapes, as well as in landscapes which are characterized by intense human activity, proved to be a combination of the analysis of archive data with the visual interpreta-

tion of remote sensing data (ALS-data, orthophotos) followed by random field checks (Proske and Bauer, 2016; Guzzetti et al., 2012).

Goetz et al. (2015) provided a comprehensive overview of different statistical prediction models for the assessment of landslide susceptibility. In our study we used the Weight of Evidence Method (WoE, first presented by Bonham-Carter (1994) with early application to landslides by Lee et al. (2004)). The main concept of WoE is based on the assumption that future landslides are very likely to occur in similar conditions as in the past (Varnes, 1984). WoE is a data-driven method representing a Bayesian approach in a log-linear form. WoE uses a prior and posterior probability for assessing the relations between (i) the spatial distribution of areas affected by landslides and (ii) the spatial distribution of analyzed landslide susceptibility factors also named predictors (van Westen et al., 2008). As a result, the degree of influence of of each predictor on past and future landslide events can be calculated.

First efforts to delineate geomorphological landforms including mass movements in the study area of Vorarlberg date back to the 1950s (Matznetter, 1956). Further approaches include those of Seijmonsbergen (1992); Aulitzky et al. (1994) and van Asselen and Seijmonsbergen (2006). In terms of landslide assessment in the province of Vorarlberg, research activities comprise shallow landslide inventories for selected large-scale test sites (Zieher et al., 2016) as well as landslide susceptibility analysis for regional-scale test sites (Schmaltz et al., 2017, 2016; Ruff and Czurda, 2008).

The government of the province of Vorarlberg published a small-scale landslide susceptibility map based on a classified geological map and the location of landslides events (details below). In order to generate a larger-scale landslide susceptibility map (including an enlarged landslide inventory), we used a routine which was already successfully applied to two other regions in Austria (Petschko et al., 2013b, 2014; Bell et al., 2013; Klingseisen and Leopold, 2006).

The only available area-wide geological map of Vorarlberg is based on information available at a scale of 1:100 000, therefore constituting a strongly generalized data basis. While geological information existing at a scale of 1:50 000 available in small parts of the province of Vorarlberg, this was not pursued further, as the different resolution of information would affect the consistency of modelling results. For instance, this rather coarse geological information does not accurately display alluvial depositions in the valley bottom, leading to inconsistencies with the DTM. In some cases, valley bottom sediments are even exceeding the limits of the foot of slopes. Consequently, the latter influences the information in the geomorphological input parameters such that e.g. the mean slope of the class *valley bottom sediments* is slightly biased.

Three major data sets provided the main input for deriving model input parameters as well as for model training and evaluation purposes:

- *Inventory of historic landslide events:* The inventory of landslide events ('Rutschungskataster') was compiled from different data sources, such as archive data provided by the Austrian Geological Survey ('Geologische Bundesanstalt', GBA), the Austrian Service for Torrent and Avalanche Control ('Forsttechnischer Dienst für Wildbach- und Lawinenverbauung', WLV), the Federal Institute for Forests ('Bundesforschungs- und Ausbildungszentrum für Wald, Naturgefahren und Landschaft', BFW), different provincial staff units, and web data mining. Events were represented by discrete points within a vector data set with one point for each database entry. The inventory of landslide events included information about the data source as well as the main hazard category and process subgroups (e.g.

slide – translational slide). Information about some 2000 gravitational mass movement events dating back to the 1930s were available (with a small number of events even earlier). We considered only approximately 800 slide-type movements for our study. Based on this subset, efforts were undertaken to map each individual landslide extent as polygon using a high-resolution LIDAR DTM with 1 meter spatial resolution in combination with natural color satellite images in order to avoid an underestimation of larger landslides. This thorough visual inspection revealed some uncertainties in the data set with respect to landslide localization, which are discussed further below. About one third of all slide events was identified and delineated this way. In addition, about 50 additional slides were discovered, delineated and included into the data set during the mapping process. For all other remaining landslides, a 50 m-buffer was assumed around the point information taken from the landslide inventory based on the average extent of slide-type events in the study area. Finally, all overlapping polygons were united, thus creating a consistent landslide mask for the study area. Note that this step also included the removal of duplicated entries resulting from the use of different data sources. The final mask layer – consisting of both mapped slides and buffered point data – contained 728 distinct landslide areas in the study area.

- *Hazard index map:* The geological map of Vorarlberg (1:100 000; Friebe, 2007) shows 171 different geological units for the entire area. To obtain suitable input parameters for landslide susceptibility modeling, these units were classified with respect to their lithological and geotechnical characteristics, generating a hazard index map (HIM). This map presented four classes which were almost identical to the main geological units. In order to avoid spurious precision, a raster cell size of 10 m was chosen when rasterizing the polygon data set. Being the least accurate data set, the hazard index map determined the spatial resolution during the entire investigation.

- *Digital terrain model:* The original DTM raster (derived from airborne laser scanning with a spatial resolution of 1 m) was used as a basis to derive several additional topographic input parameters needed, and was resampled to a 10 m grid in order to be consistent to the geogenic risk class layer available.

    - Slope: The parameter was derived by selecting the maximum rate of change in elevation value from a grid cell with respect to its eight neighbors.

    - Aspect: The value was calculated as the angle from North in degrees that features the maximum elevation gradient and classified in eight steps of 45°. A ninth class represented flat areas, which were defined to comprise steepness values smaller than 3°. These flat grid cells were excluded in the further modeling procedure.

    - Positive topographic openness: PTO characterized the widest vicinity of a raster cell (usually radial limits of 10 000 raster cells are used) to express the 'dominance' of a landscape location, giving an index of the viewshed size above the horizon line. For detailed information see Yokoyama et al. (2002).

    - Topographic position index: TPI was used to compare the elevation of a DTM raster cell to the mean elevation of its neighborhood in a radius of 100 grid cells. Positive TPI values represent locations that are higher than the average of their surroundings (ridges and hilltops), negative TPI values represent locations that are lower than their surroundings (valleys), but are highly dependent on the given radius (Guisan et al., 1999).

– Terrain ruggedness index: TRI express the amount of elevation difference between adjacent cells of a DTM. It was thus used to characterize the 'smoothness' and the very local structural heterogeneity of a surface (Riley et al., 1999).

– Topographic wetness index: TWI was used to indicate the local water availability in a cell, based on a precipitation-run off calculation. As heavy rainfall events are known to be a trigger for shallow landslide events, this parameter is also an important predictor for modelling (Bogaard and Greco, 2018; Martinović et al., 2018; Gariano et al., 2017; Guzzetti et al., 2008; Sørensen et al., 2006).

## 2.2 Traffic modeling

In order to assess the effects of road network interruptions, an agent-based traffic model was employed in the study area. The underlying data required for setting up a suitable traffic model stems from various sources. In terms of traffic services, open data provided by OpenStreetMap (OpenStreetMap Contributors, 2018), by the official road graph of Austria (GIP, 2018) and by the geodata service of the province of Vorarlberg (VoGIS, 2018) were used. The underlying road graph used for the traffic model is based on an OpenStreetMap extract, which was converted into a routeable road graph. Road capacity was derived from the functional road class. In (rare) cases of missing speed data, this information was also derived from the functional road class. Fundamental data concerning traffic demand and agent characterization are based on data provided by Statistics Austria, by the province of Vorarlberg and land use data. This includes data about e.g. traffic behavior of the local population derived from mobility surveys, socio-demographic data such as population numbers, employment statistics, or commuting flows. All of these properties can be used to model and analyze the effects of transport network interruptions on the population in the test region. The agent-based transport model which was set up on these input data was implemented in the multi-agent transport simulation framework MATSim (Horni et al., 2016). This activity-based implementation of the transport model does not only allow for large-scale agent-based transport simulations in the test area, but also retains detailed socio-demographic information on single agents represented in the model runs. The model setup implemented using aforementioned data sources constitutes the representation of traffic flow on a generic, average weekday under normal (i.e. undisturbed) network conditions. This corresponds approximately to the annual average daily traffic (AADT) for weekdays between Tuesday and Thursday. Regarding the demographic variable of employment it has to be noted that non-employment is not equivalent to unemployment. Rather, it also includes persons such as pensioners, students and home-makers.

The most significant mode of transport for the predominantly rural areas under investigation is motorized private transport, which mainly consists of cars. As such it is the mode that can be derived best from the available data and therefore is modeled in high detail on the road network. For the other modes with little available data (such as walking or bike), an origin and destination are determined, but no actual tracing of the modeled agents on the network is performed. Instead the agents simply change their position after some time ('teleportation'). For the mode public transport, a trip duration is calculated, again with no explicit simulation of mapping trips onto a transportation network. Due to reasons of relative demand coverage mentioned above, modeling car traffic can be expected to provide a good first-order approximation of the effects of circumstantial changes within the transport network for the investigated area (see section 3.3).

The full mobility simulation comprises several steps, which are facilitated by several software components. There are three main steps in conducting a full model run (Horni et al., 2016):

1. *Definition of the initial demand*: The initial demand arises from the daily activity chains of the population in the test area. It is based on the digital (routeable) road graph, points of interest along the graph and defined sequences of activities of all single agents. These *activity chains* were derived by means of population synthesis (using iterative proportional fitting) based on a mobility survey (Herry et al., 2014), socio-demographic data and land-use information. All activities of each agent are assigned to certain locations and time slots.

2. *Mobility simulation*: The actual mobility simulation is an iterative process carried out by running MATSim with a set of configuration parameters and data. It comprises the three steps of (i) *process simulation*, (ii) *scoring* and (iii) *re-planning*. Each agent features a set of plans with each plan describing the daily activity chain in the form of a desired schedule. Simulating each plan's execution allows determining associated scores, which can be interpreted as econometric utilities. The scoring function used in the simulation is the Charypar-Nagel utility function (Charypar and Nagel, 2005). This function evaluates an executed plan by considering late or early arrivals and departures (with opening hours) at facilities, costs of defined and executed activities (e.g. working, shopping, leisure time, education and habitation) and the costs of travel time[1]. Each iteration step an agent selects one plan from its set as the active one to be carried out in simulation. A fraction of all agents are allowed to optimize their score by modifying a plan of their set (e.g. in terms of trip/activity time limits 20 %, route choice 40 %, score improvement 60 %) during the re-planning step. This iterative process is repeated until the average population score stabilizes sufficiently to assume a near-optimal equilibrium.

3. *Output aggregation and analysis*: This final step comprises the aggregation of model results, which are available at a temporal resolution of 1 second in the form of event logs describing the resulting executed plans.

Following the establishment of an user equilibrium in an undisturbed traffic network state as a *baseline-scenario*, the model is re-run on the modified routing graphs for each of the landslide-scenarios considered. For each of these *incident-scenarios*, the affected network links are removed from the graph in order to indicate a network interruption caused by a landslide, and the altered behaviors of the agents – as displayed by new equilibrium states – are recorded. Consequently, the impact of network disturbances is derived by comparing the new equilibrium state behaviors on the network for any given *incident-scenarios* with the *baseline-scenario*. Generalized costs of interruptions can be obtained from this comparison in terms of e.g. affected agents, travel time or diversion lengths.

For simulation models, a trade-off between accuracy and efficiency has to be considered. Using the full population for all defined incident simulations would result in substantially longer computing times, while yielding only limited benefits in terms of explanatory power. Reduction of the sample sizes in simulation runs is a common workaround, which allows to obtain plausible estimates at reasonable computing times, while the variance of the derived results only slightly increases. Therefore, three different population samples were used for the purposes of this study:

---

[1]See Charypar and Nagel (2005) and Horni et al. (2016) for detailed information on the formulation of the Charypar-Nagel scoring function.

1. The total modeled population with roughly $2.6 \times 10^5$ persons (*baseline-100*). This population was optimized for 300 iterations to attain a solid equilibrium. It is used as a reference to determine basic properties of the population affected by the landslide incidents. The resulting activity chains were used as the basis for the following two samples.

2. A 10 % random sample drawn from all persons represented in the abovementioned baseline-scenario. It was optimized for 100 additional iterations to ascertain stability after random sampling (*baseline-10*). This sample was mainly used to establish the landslide incidents consequences' evaluation process in a less time-consuming manner, regarding that a percentage of 10 % constitutes the lower threshold recommended for MATSim to obtain consistent results (Balmer et al., 2009).

3. A 30 % random sample drawn from the baseline-scenario, sampled in the same way as the 10 % sample, again with 100 additional stabilizing post-sampling iterations (*baseline-30*). This sample is used to determine the actual consequences of the landslide incidents in a more robust manner, statistically speaking.

To mitigate an underrepresentation of traffic congestion effects, a fractional reduction in the number of agents entails that the network attributes in terms of traffic flow and storage capacity have to be reduced accordingly. For rural roads, however, with traffic volumes mostly well below maximum capacity, this is not considered to be a major issue.

## 3 Results

### 3.1 Landslide susceptibility

Visualization of landslide susceptibility derived via WoE gives a comprehensive picture of potentially critical sections of the road network in the province of Vorarlberg (Figure 2). The input data basis of events (to some extent characterized by underreporting and uncertainties regarding event localization and size) calls for careful interpretation of the resulting map. The obtained landslide susceptibility map depicts a reasonable estimate which can serve as a valuable proxy providing indications for potential road network interruptions.

The resulting landslide susceptibility maps show a consistent pattern. Valley bottoms in the Rhine Valley and the Walgau exhibit low susceptibility values, while areas characterized by Flysch rock layers (e.g. within the Bregenz Forest Mountains) are correctly identified as being particularly prone to landslides. Approximately 60 % of all landslide events show an occurrence probability of less than 40 % (Figure 3) which is considered to be moderate hazard (Neuhaeuser and Terhorst, 2007). Yet, these results are perfectly in line with our expectations, given the likely underreporting of occurred events. Therefore, it can be expected that the effective landslide risk is slightly underestimated.

Evidence from precise landslide localization and delineation proved that only about one third of all events included in the inventory could be identified unambiguously from high-resolution satellite images and a very high resolution DTM. To some extent, this is caused by imprecise information on landslide location in the landslide inventory. In some cases, evidence of landslides visible in the DTM did not coincide exactly with data points from the inventory. Rather, horizontal shifts of several

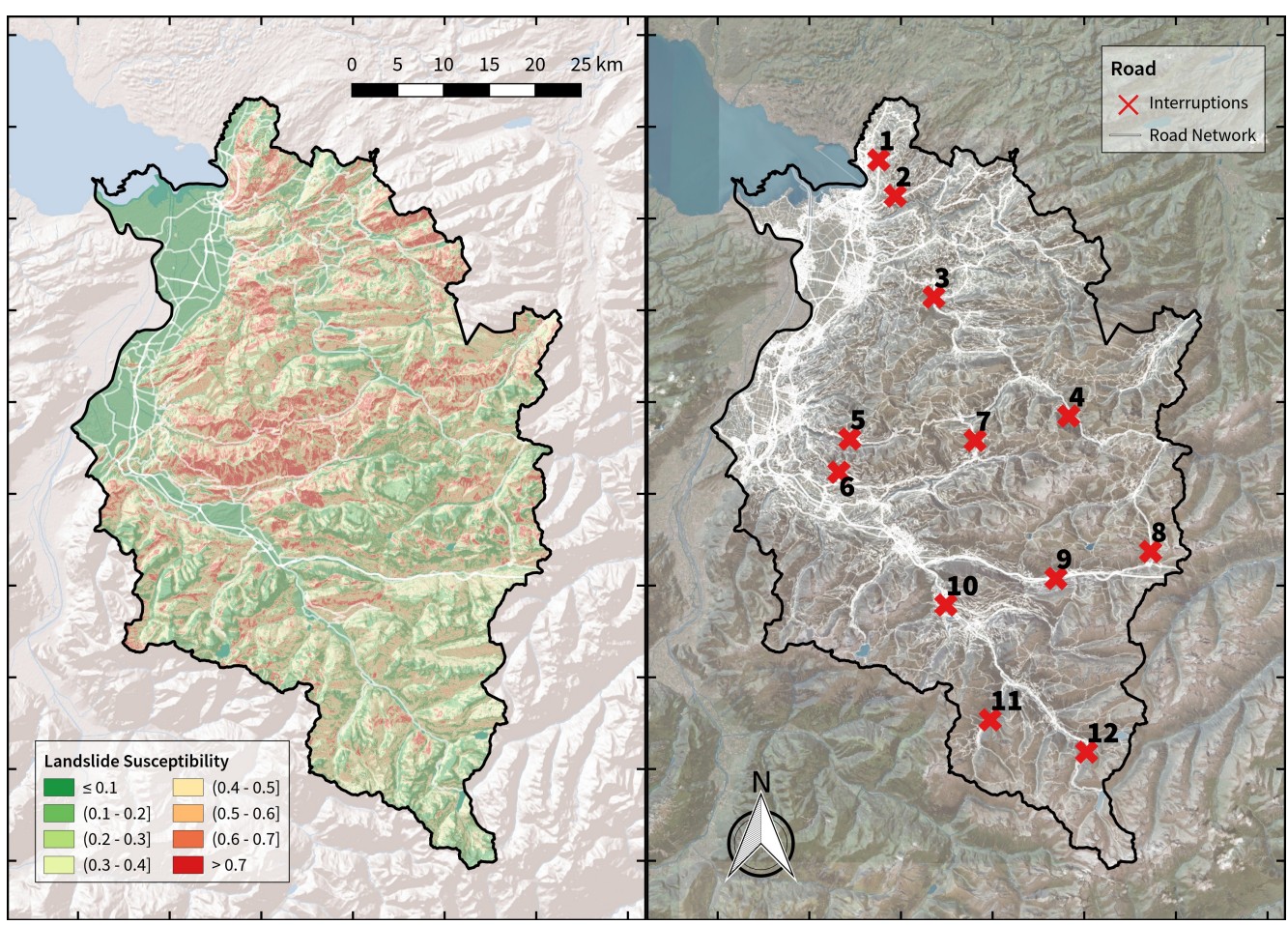

**Figure 2.** Overview of the study area featuring (a) the updated landslide susceptibility map as well as (b) the road network and the identified critical sections (incident sites).

dozen meters were noticeable, indicating inaccuracies in location information in the landslide inventory. In many other cases, no evidence of any landslides could be found in the vicinity of inventory data points. This may be attributable to the fact that small landslides affecting (agri-)cultural areas or traffic routes might be fixed efficiently, leaving no distinctly identifiable evidence.

## 3.2 Incident sites

By merging the results of the landslide susceptibility map with a digital road graph and historic data of landslide inventories, critical sections of the road network were identified (Figure 2). The selection of incident links is based on a qualitative analysis taking into account (i) landslide susceptibility, (ii) road network structure, (iii) commuting flows, (iv) extent of historic events and (v) spatial distribution of incident links within the study area. In total, twelve incident sites located in different regions of

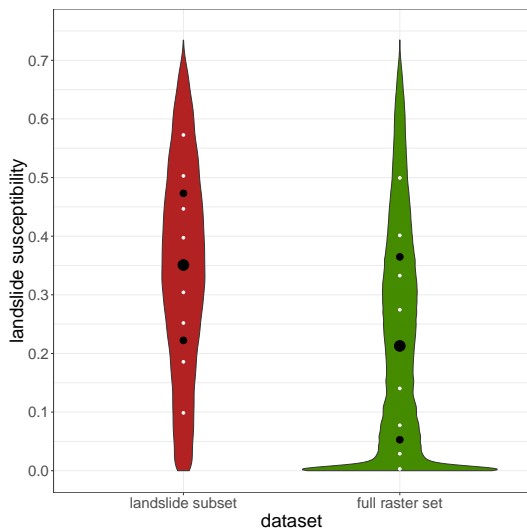

**Figure 3.** Density of landslide susceptibility values in landslide areas as well as across the whole study area. White dots indicate percentiles of the distribution (in steps of 0.1), and black dots indicate the quartiles of the distribution. Note that densities for both violin plots are displayed independently (as opposed to the conditional probability in the case of landslides) for visualization purposes. Consequently, sizes of the susceptibility density are not in the same scale and thus not directly comparable.

the case study area have been selected for further analysis of the impacts of landslide events on the road network (Table 1). Each incident comprises one or multiple links, which were flagged to indicate total blockage in case of a landslide event. This is based on the underlying assumption that occurrence probability of landslides and extent of historic events are a reasonable proxy for assuming likely network interruptions that are representative of common, everyday risks in the study area.

5    A closer look at the transport simulation model revealed that incident 12 had to be removed due to its close proximity to Silvretta-Hochalpenstraße, a high alpine toll road across the Bielerhöhe pass which connects the federal states of Vorarlberg and Tyrol. The combination of an adjacent toll road, complete road-closure during winter months and the incident's proximity to the state-border constitute a rather special traffic pattern, which is difficult to reproduce adequately in a simulation of a much larger area with significantly differing traffic patterns.

## 10    3.3    Traffic impact assessment

Transport network model simulations were conducted for the remaining eleven incident sites. For deriving insights from the agent-based transport modelling approach, we focus on the following three main aspects.

First we are interested in the transferability of the traffic model to real world traffic conditions. As the model represents traffic flow on a generic, average weekday with undisturbed network conditions it is reasonable to compare against obtainable
15    measurement data of traffic volumes, where available (VoGIS, 2018). Utilizing the baseline-scenario allows to identify the agents crossing the incident-affected links under normal conditions, as well as their properties. A direct comparison of the

**Table 1.** Overview of the twelve selected incident sites located in landslide-prone areas. The table includes the names of the affected roads, of the geographic region and the toponyms of the affected villages. The average traffic volume is displayed in terms of annual average daily traffic (AADT) and the average daily number of heavy goods vehicles (HGV) on weekdays between Tuesday and Thursday in 2016 [vehicles/day]. In addition, the peak flow per hour ($Q_{max}$) displays the maximum number of vehicles within one hour in 2016. Please note that no traffic counters are available in immediate vicinity for incidents 1 and 11 (Source: VoGIS (2018)). The last two columns contain the average and maximum landslide susceptibility value within 50 meters around the selected incident links.

| Incident | Roads | Region | Toponym | $ADT_{2016}$ | $HGV_{2016}$ | $Q_{max}$ | $WoE_{mean}$ | $WoE_{max}$ |
|---|---|---|---|---|---|---|---|---|
| 1 | L11 | Pfänder | Eichenberg | – | – | – | 0.45 | 0.71 |
| 2 | L2, L12 | Rheintal | Bregenz-Fluh | 5171 | 222 | 623 | 0.35 | 0.64 |
| 3 | L48 | Bregenzerwald | Andelsbuch | 4147 | 243 | 973 | 0.03 | 0.14 |
| 4 | B200 | Bregenzerwald | Schoppernau | 4331 | 255 | 909 | 0.17 | 0.71 |
| 5 | L51 | Laternsertal | Laterns | 1472 | 82 | 566 | 0.35 | 0.72 |
| 6 | L73 | Walgau | Dünserberg | 1693 | 287 | 263 | 0.56 | 0.72 |
| 7 | B193 | Großes Walsertal | Fontanella | 872 | 61 | 428 | 0.34 | 0.65 |
| 8 | B198 | Lechtal | Zürs | 3218 | 346 | 1048 | 0.14 | 0.54 |
| 9 | L97, S16 | Klostertal | Wald am Arlberg | 11702 | 1716 | – | 0.10 | 0.66 |
| 10 | B188, L94 | Montafon | Bartholomäberg | 14860 | 648 | 1845 | 0.24 | 0.70 |
| 11 | L192 | Montafon | Gargellen | – | – | – | 0.14 | 0.44 |
| 12 | B188 | Montafon | Partenen | 3217 | 236 | 546 | 0.17 | 0.48 |

annual average daily traffic on weekdays between Tuesday and Thursday (Table 1) with the simulated number of affected car trips (Table 2) shows considerable variability depending on the location of incidents and the affected road class (e.g. highway, secondary road). While most of the incidents are located in rural areas or at roads with low daily traffic, some of them are very close to (semi-)urban regions or along main roads. Incident 10 is on the main road at the entrance to the Montafon region and therefore an essential part of the road network. This is reflected by the huge number of affected agents and car trips, which are modeled rather fittingly. Incident 11, on the other hand, is at a road segment of a valley's head with few affected agents (Table 2). All incident links providing access to skiing-resorts (e.g. sites 8, 9) can be expected to deviate strongly, as there was no data on tourism-induced traffic available in the mobility survey that served as basis for the traffic model. In some cases the simulation will choose to guide traffic flows on alternate routes (e.g. shift from site 6 to 5 and 7) which are similar with respect to functional road class and travel time. Additional considerations are explored in the discussion below (see section 4.2).

Second we outline the major features of the population affected by each road network obstruction caused by landslides. This also is information gained from analyzing the baseline scenario of the model. Table 2 shows selected incident sites with a broad variability in terms of average daily crossings, ranging from as little as 118 up to more than 16 000 daily car trips. Considering the results of the baseline scenario, both median duration and median distance of daily car trips as well as the share of mode car indicate a strong reliance on this transport mode. Depending on the location of the incident blockage points, about 50 – 80 % of

**Table 2.** Summary of car trip characteristics for agents in the traffic model of the undisturbed baseline scenario, to be affected by incident scenarios. Affected agents refer to those crossing the incident site at least once, within their regular daily plans. Employment rate is the share of working people within this group. Affected car trips designate the number of incident site traversals by those agents in that transport mode. The share of mode car gives the proportion of site traversals by car relative to all traversals mentioned before (in any mode). Site traversals of incident links by car undertaken by employed people are shown as percentage in column six. Medians of daily travel time and distance are displayed for employed and not employed people, respectively, which again refer to all affected agents and their trips crossing the respective site.

| Incident | affected agents [] | employment rate [%] | affected car trips [] | share of mode car $\cong$ [%] | car trips by employed [%] | median daily travel time [h:m:s] | | median daily travel distance [km] | |
|---|---|---|---|---|---|---|---|---|---|
| | | | | | | employed | not employed | employed | not employed |
| 1 | 565 | 83.54 | 1363 | 93.81 | 82.17 | 00:46:58 | 00:51:13 | 46.4 | 54.9 |
| 2 | 1486 | 64.20 | 3968 | 91.66 | 63.38 | 01:01:02 | 01:12:18 | 62.5 | 76.4 |
| 3 | 4794 | 64.15 | 13856 | 93.15 | 62.12 | 01:31:14 | 01:43:30 | 91.5 | 106.3 |
| 4 | 586 | 58.42 | 4285 | 93.33 | 56.24 | 02:07:49 | 02:22:19 | 118.2 | 143.5 |
| 5 | 858 | 67.53 | 3032 | 95.74 | 63.36 | 01:18:40 | 01:47:11 | 79.3 | 110.9 |
| 6 | 128 | 73.19 | 355 | 89.65 | 71.27 | 00:53:31 | 01:06:54 | 58.3 | 68.9 |
| 7 | 572 | 63.68 | 3200 | 93.73 | 60.81 | 01:52:53 | 01:59:40 | 103.2 | 118.8 |
| 8 | 1404 | 49.68 | 6305 | 92.93 | 49.72 | 02:13:48 | 02:18:50 | 159.4 | 166.8 |
| 9 | 576 | 52.06 | 7815 | 92.07 | 51.63 | 02:01:21 | 02:12:20 | 149.1 | 160.6 |
| 10 | 4709 | 52.16 | 16372 | 91.87 | 51.62 | 01:24:32 | 01:32:02 | 104.4 | 122.1 |
| 11 | 5 | 74.00 | 118 | 87.41 | 75.42 | 01:14:50 | 02:08:56 | 75.2 | 156.5 |

all trips were attributable to working persons. A consistently similar percentage was found for the employment rate of affected agents, allowing the conclusion of similar stratification regarding the variable of employment, which barely influences the modal share of car-use. The ratio of affected agents to car trips was mostly around 3.5. Cases 9 and 11 show strong deviations. According to the medians of baseline results, which were used as robust indicators for comparing distributions, non-employed persons complete longer daily trips with respect to both distance and time.

As exemplary properties, age and employment status of agents within the total synthesized population affected by each incident are shown in Figure 4. The age range tails are trimmed to the data range. The widths of the distributions are scaled to the absolute agent counts, therefore displaying relative numbers of affected individuals (see Table 1). Employment status was distinctly related to a specific age range.

As a third aspect, differential changes in travel patterns within the model were assessed. They can be analyzed by comparing the newly established traffic equilibria, which result from the interruptions occurring at the defined incident sites, against the baseline situation. In some situations the blockage of a non-redundant link can occur, meaning that no alternative routes are available, as is the case for incident 11. Here, it is of no benefit to run a traffic simulation on the modified road graph affected by the landslide event. All agents striving to cross the incident site would simply be marooned in a valley or be unable to reach it by car from the outside, respectively. In addition, there would be minor gains for all other road users due to slightly

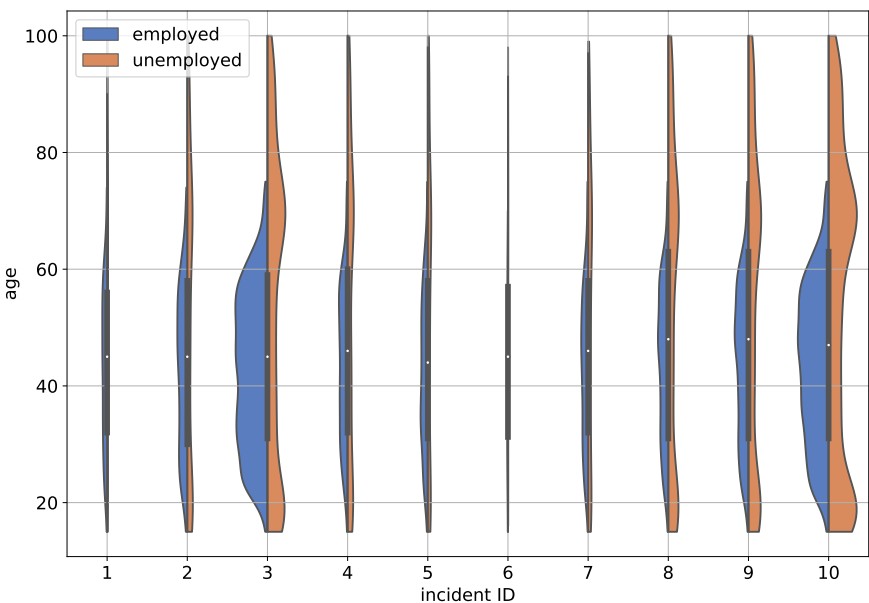

**Figure 4.** Kernel density estimates of the number of agents by age and employment status of agents affected by each incident.

less traffic on the network. Results for all other incidents show a broad range of possible scenarios (see Table 3) that might occur in case of network interruptions. The *detour length*, being the shortest alternate route length between the ends of the road links that were severed by the landslide, ranges from around 3 km up to 108 km. This distance is not necessarily reflected in the *diversion lengths* of actual car trips made in the simulation, because agents learn to give these interruptions an adequately

wide berth, still striving to optimize their score. While optimized circumnavigation results in an increase of travel time across the board (*diversion time* > 0), our findings show that travel distance might actually decrease in some cases (*diversion length* < 0). This is attributable to the choice of lower priority alternate routes with smaller effective capacities (due to road class and/or congestion) resulting in lower traveling speeds, yet shorter lengths. The quartiles were chosen to convey an impression of the distributions underlying the aggregated diversion quantities. Given the geographic situation in the study area, the closing

of specific road segments often results in a considerable increase of travel distances. This is illustrated by the examples of incidents 4, 8 and 9 where detour length exceeds 100 km as there are no parallel routes traversable at comparable utility score in the obstructed road network model.

    Figure 5 shows relative diversion quantities, which were obtained by comparing the daily aggregates of every agent's trips for each incident scenario against their counterparts in the baseline scenario. A summary over all affected agents across all

15 incident scenarios and both travel time and distance shows the median of relative quantities to increase by approximately 7 %. Results show that up to two times the amount of initial travel time and distance is required for evading a majority of the modeled incidents. In some outlier cases, up to the sevenfold time and distance is spent making the agents' daily journeys by car. However, on exceptionally rare occasions (i.e. 3 out of 5 518 agents affected by incidents in the 30 % sample) it can also be

**Table 3.** Summary statistics of differences between each interruption scenario and the undisturbed baseline scenario, for both (10 % and 30 %) population samples. They are expressed in terms of quartiles of (additional) diversion lengths and times. For incident site 10, two individual stretches of road are affected (NE/SW), therefore two different detour lengths are indicated.

| Incident | detour length [km] | incident scenarios: 10 % sample | | | | | | incident scenarios: 30 % sample | | | | | |
| | | diversion time [h:m:s] | | | diversion length [km] | | | diversion time [h:m:s] | | | diversion length [km] | | |
| | | $q_{25}$ | $q_{50}$ | $q_{75}$ | $q_{25}$ | $q_{50}$ | $q_{75}$ | $q_{25}$ | $q_{50}$ | $q_{75}$ | $q_{25}$ | $q_{50}$ | $q_{75}$ |
|---|---|---|---|---|---|---|---|---|---|---|---|---|---|
| 1 | 16.4 | 00:05:41 | 00:12:10 | 00:22:34 | 5.04 | 8.18 | 13.01 | 00:08:07 | 00:12:02 | 00:23:34 | 6.27 | 8.94 | 17.01 |
| 2 | 50.2 | 00:14:47 | 00:31:48 | 00:52:06 | 14.71 | 21.80 | 46.10 | 00:15:53 | 00:25:14 | 00:44:55 | 12.00 | 20.92 | 38.81 |
| 3 | 11.9 | 00:04:05 | 00:07:02 | 00:10:21 | 7.28 | 9.79 | 17.06 | 00:03:20 | 00:06:15 | 00:08:53/ | 7.19 | 9.57 | 17.30 |
| 4 | 106.9 | 00:06:32 | 00:16:11 | 00:44:30 | 37.85 | 61.17 | 72.97 | 00:09:23 | 00:19:43 | 00:52:20 | 39.47 | 66.26 | 86.44 |
| 5 | 65.7 | 00:12:51 | 00:16:10 | 00:49:54 | 7.63 | 19.88 | 34.52 | 00:10:58 | 00:24:44 | 00:51:10 | 11.09 | 22.90 | 45.14 |
| 6 | 27.2 | 00:07:56 | 00:10:36 | 00:11:25 | 11.90 | 15.17 | 15.89 | 00:04:20 | 00:05:43 | 00:11:20 | 7.11 | 8.28 | 15.21 |
| 7 | 65.7 | 00:06:24 | 00:13:22 | 00:21:22 | 18.32 | 27.39 | 46.30 | 00:06:52 | 00:11:52 | 00:21:32 | 18.61 | 32.58 | 48.13 |
| 8 | 107.9 | 00:10:42 | 00:21:18 | 00:34:51 | -30.52 | -15.82 | 8.61 | 00:13:09 | 00:27:58 | 00:42:52 | -22.90 | -5.38 | 16.51 |
| 9 | 107.7 | 00:15:05 | 00:31:50 | 01:14:09 | -21.62 | -0.71 | 31.43 | 00:17:15 | 00:35:56 | 01:15:42 | -17.98 | 3.64 | 37.22 |
| 10 | 9.6 / 2.6 | 00:02:38 | 00:04:01 | 00:05:31 | 0.79 | 1.42 | 1.79 | 00:02:30 | 00:03:10 | 00:04:16 | 0.84 | 1.68 | 1.77 |

observed that daily travel time and distance decrease marginally ($\approx -0.2\%$) relative to the baseline scenario. This seemingly paradoxical phenomenon is explored further in the discussion. An obvious feature when comparing the employment status are the wider tails in the distributions of diversion costs for employed drivers, as indicated by taller boxes and longer whiskers. This is attributable peak traffic conditions due to occupational time constraints resulting in wider detour variations.

Utilizing the disaggregated population information retained in the data, we can investigate the spatial distribution of distinct consequences on the agents. In Figure 6 the agents' home locations are used as spatial references. The residences of affected persons are distributed differently for the various incidents, naturally. However, due to the mountainous landscape with habitation predominantly located in the valleys, incidents can have far reaching impacts for commuters who usually traverse the main traffic axes along those valleys. While the home location was chosen as a spatial reference in the shown visualization,

diversion times and distances can be spatially related to any other location connected to an agent's activity chain (e.g. main activity location, workplace location).

## 4   Discussion

This study examined the different impacts of landslide-induced road network interruptions on communities in a rural area in the European Alps. In general, road network systems are highly complex – most notably regarding their socio-economic

effects on communities. Alpine areas are particularly vulnerable to such interruptions and changes. Firstly, this is based on their topography. Mountain regions are usually prone towards natural hazards events, often lacking the feasibility to build a redundant transport system. Secondly, because of their socio-economic properties, mountain regions are strongly dependent on

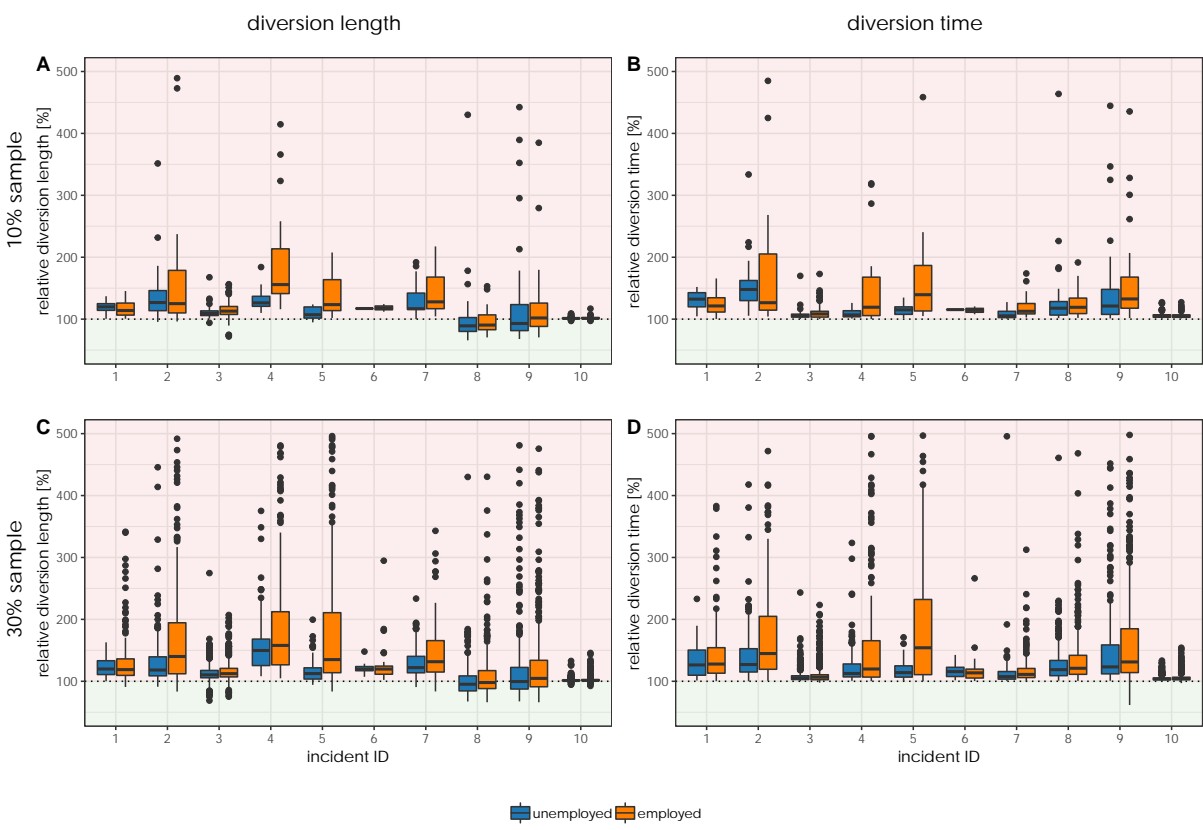

**Figure 5.** Relative changes of diversion length and diversion time with respect to the baseline scenario, resulting from landslide-induced network impairment. Boxplots in the upper / lower row show results for the 10 / 30 % sample, respectively. A discrimination of diversion costs between employed and non-employed agents is color-coded. Note that the y-axis has been limited at 500 % for better readability. Occasional outliers up to 700 % however do occur.

the established road network (i.e. dependency on tourism and high numbers of commuters). This implicates a political debate on the question of how to manage mountain road network systems.

## 4.1 Landslide modeling

A WoE approach relying on occurred events and various geophysical properties as input data was applied in order to obtain a 5 refined landslide susceptibility map for the study area.

Results of the landslide susceptibility assessment showed a clear quality improvement over preceding efforts, which was mainly based on geological data. The inclusion of additional input parameters as well as an increased spatial resolution provided added value in comparison to the currently available governmental landslide susceptibility map ('Gefahrenhinweiskarte') of Vorarlberg. However, efforts undertaken within the modelling procedure revealed some shortcomings in the data basis.

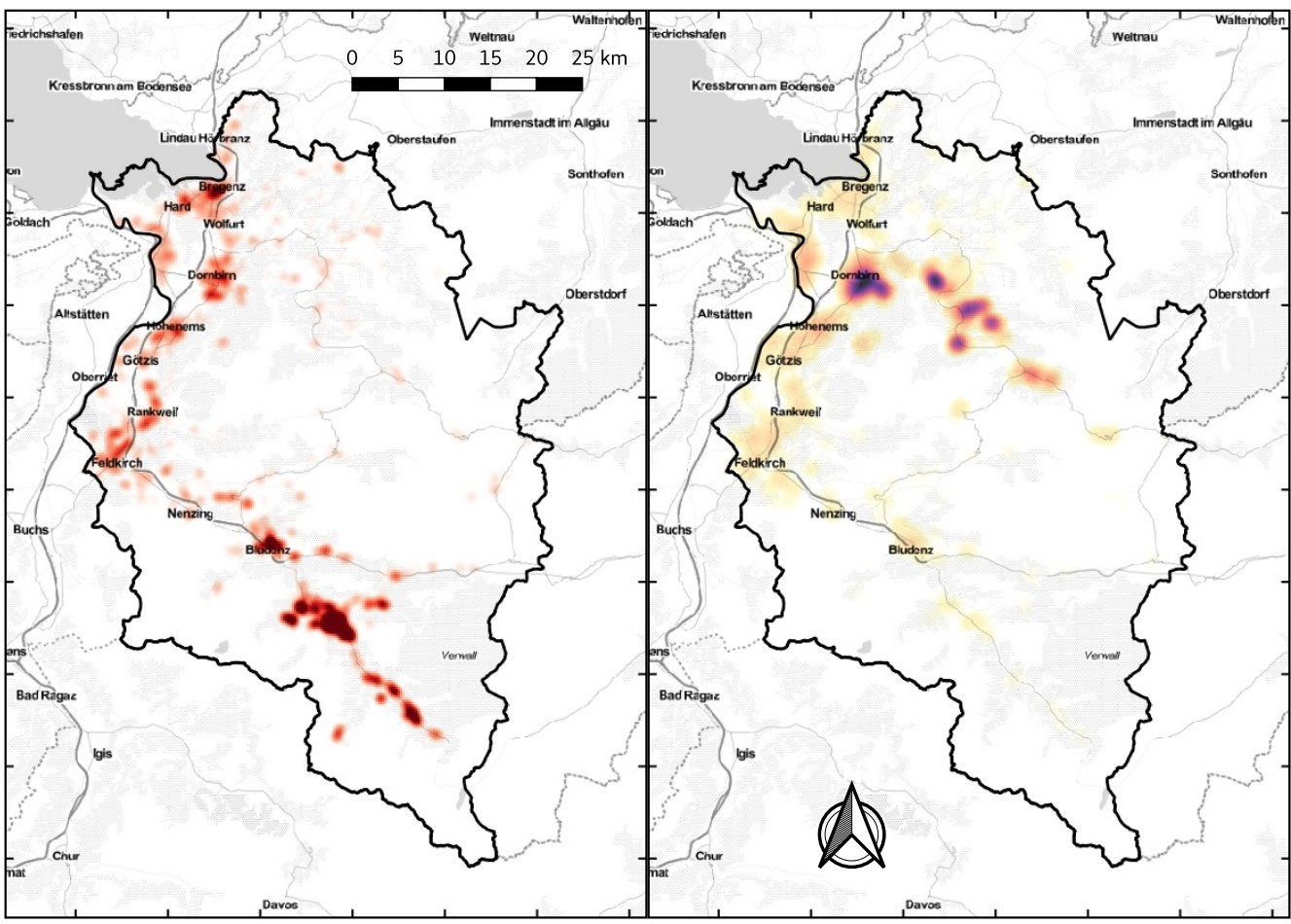

**Figure 6.** Spatial distributions of home locations of persons affected by incidents (based on the 30 % sample). Left: Home location density of affected persons for incident 10 (darker means higher density). Right: Relative diversion time for incident 3 (delay over total daily travel time – c.f. Figure 5) inscribed at each person's home location (darker means greater aggregated diversion times).

Both an accurate landslide inventory and precise geological information at a high spatial resolution substantially supported the modeling approach, as already successfully shown for other provinces of Austria, such as Lower Austria (Petschko et al., 2014), Styria (Proske and Bauer, 2016) and Burgenland (Leopold et al., 2017).

In terms of spatial consistency of the landslide inventory data, events were not equally distributed across the study area. This is reflected by obvious disparities in data density, which are attributable to different perception and documentation in different municipalities. Landslide events were either mapped spatially distributed (e.g. Blons, St. Gerold) or aggregated to spatial event clusters (e.g. Laterns, Fontanella). In some municipalities, significant underreporing of events (compared to neighboring municipalities) was evident. Hence, municipalities within the same geological sub-units showed substantially different results in the quality of landslide reporting.

In addition, landslide events were only represented as point data inventory, including only an additional tag referring to location accuracy ('exact' vs 'inexact') without any consistent additional information on landslide localization or extent. Literature research and archive data were cited as main data sources for data acquisition, while sources for several events (approx. 15 %) were entirely missing in the dataset. Hence, acquisition and reporting of events did not seem to be performed in a consistent way across the whole province.

An area-wide mapping of landslides (i.e. delineating polygons of landslide extent based on very high resolution DTMs and optical satellite images) is considered to be a crucial data base for modeling purposes (Zieher et al., 2016; Petschko et al., 2013b). Even though we established such a dataset, leading to a quality improvement of the landslide susceptibility map, there is still room for progress. Landslide verification by means of visual interpretation of DTMs and ortho-images rarely shows clear evidence of landslides (Petschko et al., 2015). Main issues identified during the modeling procedure were related to inventory incompleteness (e.g. underreporting, especially of smaller events which may be fixed quickly by local authorities or farmers), biases (spatial distribution of events, e.g. across administrative boundaries) and inaccurate localization (including the extent). In order to address these issues, a substantial amount of resources (for the reconstruction of historic events using multiple archive data and a subsequent mapping of all events using high-resolution geo-data) would be required. The extent of these efforts has recently been illustrated by Schmaltz et al. (2017), who built up a comprehensive inventory of events for a comparably small area within the province of Vorarlberg.

At the same time, the landslide mapping process revealed the importance of an accurate polygon delineation when working on large scales. Our findings showed that landslide events recorded in the inventory are not located in the centroid of landslide areas, but rather at the specific location where damage was reported (e.g. at landslide scarp or toe).

## 4.2 Traffic impact assessment

Regarding the application of an agent-based traffic model for assessing the impacts of landslide-triggered road network interruptions in a rural alpine area, our findings highlighted several issues:

Generally speaking, medians of total daily car travel times and distances appeared to be rather long (Table 1), which is a result of the topography in the study area and the related limitations in spatial planning alternatives. Agglomerations are mainly located either in the Rhine valley or its tributary valley, the Walgau, which results in long driving routes for commuters inhabiting remote areas. Furthermore, winter sports centers were found to be slightly underrepresented n the results for two reasons: Firstly, only data about the local population was available from mobility surveys. Secondly, a generic average weekday was modelled, thus seasonal effects were levelled out.

It also can be observed that total daily trips of employed agents were shorter than those of the non-employed – as far as both daily travel time and daily travel distance are concerned (Table 1). This can be explained by the daily schedules of the non-employed agents who fulfill various chores which shows in the consistently smaller percentage of car trips by employed agents compared to the employment rate. The occasional extra car-trip by non-employed agents resulted in daily travel differences of approximately 5 – 10 %.

Concerning diversion effects as shown in Table 2, diversion times were sometimes surprising when paired with the corresponding quartile of diversion length. A small increase in travel time can still be accompanied by a larger increase in travel distance, given the choice of an appropriate high speed (and capacity) road. For incidents with shorter detour length (Table 2) the inter-quartile range ($q_{75} - q_{25}$) of diversion times tends to be reduced in the 30 % sample when compared to the 10 % sample, while for the incidents with long detours (i.e. incident 4, 8 and 9) there is an opposite tendency. The source of these small effects is yet to be explored in detail, but it is most likely due to the higher number of agents required to spread out over a wider area of the network, thus causing network loading effects in the simulation which are not fully compensated by the population-equivalent scaling of road capacities.

Our results indicate that employed agents in general were more affected by network interruptions, for the medians of relative diversion costs being mostly higher (Figure 5). As it is the case with the wider-tailed distributions of diversion cost for employed agents, this is due to stricter time constraints in terms of working hours, entailing that the employed agents are more prone to congestion effects caused by rush hour traffic peaks.

A somewhat paradoxical situation emerged at incidents 8 and 9, where results show negative diversion lengths compared to the baseline scenario. Such effects occurred when agents were forced to use roads featuring a lower functional road class (due to lower speed limits, or at least lower effective speeds) than in the baseline scenario, thus entailing shorter travel distances but longer travel times. For instance, instead of using the highway S16 to cross incident link 9, agents had to use rural roads between the two corresponding highway junctions, which led to a decrease in travel distance at the expense of a corresponding increase in travel time.

Even though diversion times show consistent overall increases, there were sporadic cases where both travel time and distance decreased in comparison to the baseline scenario. These cases, however, were extremely rare. The establishment of a new user-equilibrium on the road network can – for very few agents – sometimes result in better individual outcomes under worsened overall conditions.

The models for the disturbed situations showed shifts in agent trip start and end time. This indicates that agents naturally learn to e.g. adjust their departure time to reach their offices in time. However, no general patterns can be derived from these observations, as these adjustments vary strongly between individual agents.

However, results have to be interpreted under consideration of several limitations. Because the points of interest within the daily plans of agents were static, the re-planning procedure so far did not include re-assignment to substitute facilities. This may not appropriately reflect reality since it is very likely that – in case of activities where alternative facilities exist – people would adjust their diversion routes accordingly, such, as, for example, shifting shopping activities to better accessible locations.

For the employed traffic model, the reasons for deviations from traffic measurement data were the following: Firstly, our model only considers trips by inhabitants of the study area. Consequentially, cross-border traffic from both people commuting from outside (e.g. Tyrol, Germany, Switzerland) to Vorarlberg as well as transit through Austria are not taken into consideration. While this restriction has less effects in rural areas (where many of the considered incidents were located), effects will get more pronounced in areas next to a (national) border or on highways with high volumes of transit traffic (see Table 1). Secondly, distortions may arise due to an unavoidable temporal mis-alignment of utilized data sources. While the underlying mobility

household survey that serves as basis for the MATsim model was conducted in 2013 (Herry et al., 2014), the available traffic counts refer to the year 2016.

## 5   Conclusions

In this paper we have shown that agent-based traffic modeling allows to gain interesting insights into the impacts of road network interruptions on the mobility behavior of affected communities by modeling their responses to network disturbances. The detailed representation of single agents in the transport model allows for optimizing certain characteristics of agents (e.g. time of departure, route choice, activity list, etc.). Generalized costs of interruptions (i.e. monetary costs, time losses, etc.) can be obtained by employing a utility function to the agents' resulting behavior.

Choosing a feature-enriched description of the basic population within the area of interest – as it becomes possible by use of increasingly fine-grained data over the recent years – allows to derive a very detailed picture of changes effected by systemic disturbances. These details can be any combination of socio-demographic or spatio-temporal characteristics which pave the way for more precise decision making and implementation of guiding measures. Therefore, the MATSim implementation is considered to be suitable for providing an agent-based analysis of expected impacts on changes in the traffic system.

Our findings may also provide a basis for future work in this area, which should expand the limits of the present study by incorporating transit and cross-border traffic and might shed more light on traffic displacement effects. Moreover, further work could be devoted to the economic analysis of interruption costs.

While this study has explored road network vulnerability against the background of landslide susceptibility, the presented methodology is easily transferable to other (natural) hazards that might cause network interruptions, such as e.g. avalanches, floods, or terrorist attacks.

Despite some limitations, the efforts undertaken in this study can offer valuable guidance for hazard managers and decision makers by providing a sound estimation of likely implications of landslide-triggered road network interruptions on local communities. In particular, information on the number of affected people, their employment status as well as their associated costs (in terms of both time loss and diversion length) can serve as a basis for gauging possible adaptation and protection measures in a broader context. Additionally, such findings can contribute to decision making by prioritizing such measures in line with budgetary constraints. Supplying decision support on where and how to efficiently and effectively allocate limited resources is beneficial, and it enables tackling the impacts of adverse weather events and natural hazards by means of appropriate measures.

*Data availability.* Special emphasis was put on using open data and open source software wherever possible. All openly accessible data sets used are listed in the following: The official road graph of Austria is available via the Austrian Graph Integration Platform GIP at http://gip.gv.at/ (GIP, 2018). Additional geodata can be found at the geographic information system of the federal government of Vorarlberg, accessible via http://vogis.cnv.at/ (VoGIS, 2018). The governmental landslide susceptibility map as well as historic event data can be accessed through the HORA (Natural Hazard Overview and Risk Assessment Austria) platform at http://www.hora.gv.at/ (eHORA, 2018). The routing

graph is based on an OSM data extract from https://download.geofabrik.de/europe.html (OpenStreetMap Contributors, 2018). The landslide susceptibility map is available as a supplement to this paper.

Apart from the traffic simulation, which was implemented in MATSim (Horni et al., 2016), all data preparation, processing and visualization has been done in **R** (R Core Team, 2018) using the tidyverse framework (Wickham et al., 2018), in Python (Python Core Developers, 2018) with its scientific tool stack SciPy (Jones et al., 2018), and in QGIS (QGIS Development Team, 2018).

*Competing interests.* Sven Fuchs is a member of the Editorial Board of Natural Hazards and Earth System Sciences. Otherwise, the authors declare that they have no conflict of interest.

*Acknowledgements.* We would like to thank the *Austrian Federal Ministry for Sustainability and Tourism* (BMNT) and the *Land-, forst- und wasserwirtschaftliches Rechenzentrum* (LFRZ) for providing basic hazard maps as well as data on landslide events. We are grateful to the provincial government of the federal state of Vorarlberg for providing the 1 m DTM and the geological map. We also appreciate the valuable input of Melitta Dragaschnig (AIT) and her assistance in obtaining the OSM graph. We thank Melina Frießenbichler and Clemens Moser (ZAMG) for their assistance to landslide mapping.

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

**Supplement 1: Interactive map 1**

Interactive map (Leaflet) of the study area, including the calculated landslide susceptibility and the location of the incident sites over several different basemaps.

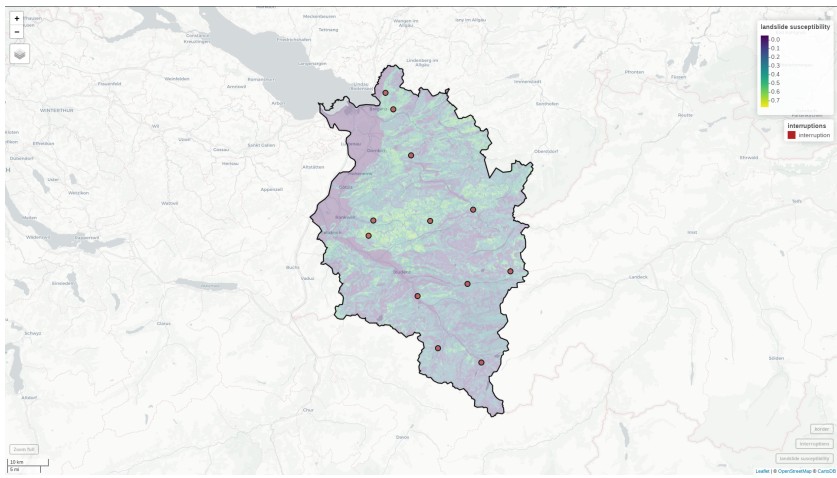

**Supplement 2: Interactive map 2**

5   Interactive map (Leaflet) of the study area, showing the home locations of all agents affected by any incident (based on the 30 % sample) over several different basemaps.

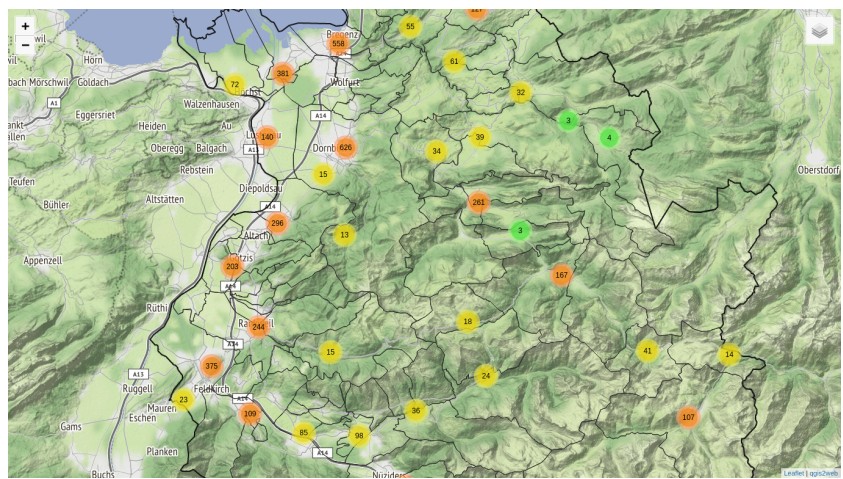