# Peer review of "On the nexus between landslide susceptibility and transport infrastructure – an agent-based approach"

_Natural Hazards and Earth System Sciences, 2018_

## Referee Comment (RC1) · Anonymous Referee #1 · 11 May 2018

**1 General comments**

The authors describe a method to study the impacts of landslides on the travel behaviour of road users in a mountain area in Austria using a landslide susceptibility map and an agent-based traffic model. Clearly, a lot of thoughts and efforts were used to set up the models and do the analysis.

Although the paper provides some interesting insights on the investigation of transport infrastructures prone to landslides, there are some critical points that should be addressed before consideration for publication.

1. In the tile, the paper implies an ". . . agent-based vulnerability assessment of rural road networks . . . ". Unfortunately, this is not delivered, i.e. at the end of the paper, the reader does not have a clear answer on the question "How vulnerable is the road network (in Vorarlberg) towards landslide hazards?" Instead, 10 (very specific) Scenarios were analyzed and compared, considering the detour length and the evasion time. Hence, the concept how to assess network vulnerability has to be better expressed, beginning with a clear definition how the authors define vulnerability in their paper, and the introduction of some vulnerability measures for quantification and justification (e.g. using indices, curves, tables, maps, etc.).

2. Throughout the introduction, several times the importance of socio-economic impacts and the severe losses caused by the disrupting services are mentioned. Finally, only the prolongation of the travel is considered. Since the authors already implemented a very detailed agent-based model with many socio-demographic data (e.g. the agents are employed/unemployed), why not actually assessing the socio-economic impacts? This would also be a novel contribution of the paper, which is currently missing.

3. In the current version of the paper, the methodology can be summarized as: "Running a traffic model, thereby disabling a selected link", which is not a novel concept, or method. What's missing is a clear description of the novelty for the proposed methodology. i.e. What is new? How is it better than other approaches?

4. Concerning the landslide susceptibility map:

(a) There is a conceptual flaw, using landslide susceptibility maps for assessing network-related processes. Contrary to building assets (e.g. houses, facilities, etc.), networks are used to describe dynamic processes (e.g. traffic flow), with the consequence that local events can have a severe impact on the whole network (as the authors showed in their example). The problem is that landslide susceptibility maps describe only the relative likelihood of future landslides, however, since the network is more than its components, there is a probability that a very unlikely landslide (low susceptibility), causes more harm than a very likely landslide. For example, there is a very low susceptibility that a landslide will be triggered and affecting a major connection, causing thousands of people to stay at home while incident 6 affects only 128 agents. At the current state of the paper, such network effects are completely neglected, however, this is the core concept (and challenge) of analysing network structures.

(b) Why did the authors develop a landslide susceptibility map although "The government of the province of Vorarlberg offers an official landslide susceptibility map . . . " (Page 5/ Line 26) and "The official hazard map already provides a reasonably accurate and consistent basis for the purpose of identifying vulnerable sections." (11/30)? Additionally, the used Weight of Evidence Method (Bonham-Carter, 1994) is nothing new, and therefore worth to spent 4 pages of the paper, only to figure out that the official landslide susceptibility map is a good enough estimate.

5. Concerning the selection process of links to be blocked

(a) It is absolutely not clear, how the 12 incident sides were selected. Please, give a detailed description how this was done (quantitative?, qualitative?), especially since the authors remove later on selected sides (". . . 12 had to be removed due to its close proximity to Silvretta-Hochalpenstraße . . . "(13/1).)

(b) Also, it would be of interest which landslide susceptibility is associated with each incident side. See point 5: If only areas with high susceptibility are considered, the question arises, if there is not a scenario where the road network is more vulnerable to landslides in less susceptible areas.

(c) An important part missing is the interaction between landslides and road network. In the current version, it is assumed that a landslide occurs at the incident side and completely damage (block) the road section over a period of at least 24 hours (runtime for the MATSim model)? If so, these are very strict assumptions and is contradicted by the author's statement "...due to the fact that landslides, which affected traffic routes or (agri-)cultural areas, are usually fixed quickly and efficiently." (11/10). Also, how could such assumptions be made without the knowledge of the particular landslide type (initiation and run-out, volume, speed)? The likelihood of the occurrence of landslides is not a sufficient reason to assume a damaged infrastructure. Please, specify the assumptions made and give a detailed description how the (physical) damage of the infrastructure was derived from a landslide susceptibility map.

6. Concerning the agent-based traffic model

(a) The implementation of an agent-based model is very ambiguous, please clearly state why such an approach was used, especially since most of the results (affected persons, detour lengths, evasion times) could also be observed by a flow-based traffic assignment.

(b) In the current version of the paper, several assumptions made and several limitations of the traffic model are not clearly stated. e.g. the MATSim simulation considers only/maximum one day, an agent has perfect knowledge of the interrupted section, origin and destination do not change during and after extreme events, etc.

(c) A major shortcoming is that only trips of inhabitants of Vorarlberg are considered, which does not reflect reality and certainly leads to an underestimation of the socio-economic impacts in the region. The question is how can the vulnerability assess given this constraint? Additionally, how could the traffic model be calibrated and validated, neglecting a majority of the travellers on the network?

(d) Why was so much focus put on introducing and analyzing 10% scenarios, without any additional benefit for doing so? It could have been stated, that for computational reasons a pre-sampling with 10% of the agents has been done, but the evaluation has been done with a 30% scenario.

(e) It is not clear how many simulations (not iterations) have been done for each incident. In other words, how often was the traffic model run for one incident? Since the agent-based model tries to optimize the behaviour of multiple agents, the simulation results might change over time.

(f) Using advanced modelling tools often suggests precise outcomes, however, since many unknown input parameters are necessary, the results might come with high uncertainties. These uncertainties have to be quantified in order to make meaningful statements. At least a more detailed (quantitative) description how accurate the traffic model compared to the actually measured traffic volumes should be given.

7. As mentioned in the beginning, it is hard to interpret the results and conclude how vulnerable the road infrastructure is. For example for side incident 10, 4709 agents are affected by an average evasion time of 3:10 minutes over a whole day. Does this mean there is almost no vulnerability against landslides? How can road authorities derive conclusions from this results? Should they invest in some protection measures or not?

8. Given the shortcomings of the current version, I kindly would like to suggest a rejection and encourage the authors to re-submit once the major issues mentioned

above are fixed and the storyline is clear and focused on what has been promised in the title.

**2 Specific comments/questions**

1. (1/10): "The focus of this case study is on resilience issues and support for decision making in the context of a large-scale sectoral approach." this is clearly not the case in this paper. Either this will be added to the paper of this statement should be deleted.

2. (1/15) Here only single events are considered, however, in reality, we often have to deal with the occurrence multiple hazard events (e.g. heavy rainfall caused several landslides). How can the proposed methodology cope with such situations?

3. (7/23) "Road capacity was derived from the functional road class." For personal interest, how was this done and in which range where those values per road class?

4. (10/1) Figure 1. The different road classes should be indicated (e.g. highway, primary road, ...) in order to give the reader an overview how the network is structured and where the major links are located. Additionally, since the base scenario is already computed, a map with the traffic volume should be added, to indicate the traffic flow. Next, to such a figure, it would be interesting to see a figure for the traffic volumes of an interrupted network.

5. (14/3) "In some situations, the blockage of a non-redundant link can occur, meaning that no alternative routes are available, as is the case for incident 11. Here, it is of no benefit to run a traffic simulation on the modified road graph affected

by the landslide event." Actually, what would happen is that the overall travel time will decrease for the network since fewer people are on the roads. The issue of missed trips (people who are cut off from the network) is neglected in the current version of the paper, however, it is important problem and should also be treated. Especially since this could cause more socio-economic impacts than a trip prolongation of several minutes.

6. (15/9) How many agents were simulated? 30% of 260000 is 78000 and not 5518. Probably this sentence has to be clarified.

7. (16/1) Figure 2. Why showing the 10% and the 30% example, is there any additional value in showing and discussing the 10% example?

8. (19/4) "In this paper, we have shown that agent-based traffic modelling allows gaining interesting insights into the impacts of road network interruptions on the mobility behaviour of affected communities by modelling their responses to network disturbances." This might be true but is only slightly related to the topic of road network vulnerability which was promised in the title of the paper.

**3  NHESS aspects**

1. Does the paper address relevant scientific and/or technical questions within the scope of NHESS?

Yes, the paper tries to addresses the question of vulnerability related to networks exposed to landslides.

1. Does the paper present new data and/or novel concepts, ideas, tools, methods or results?

Not in the current state. Right now the method is "Running a traffic model, thereby disabling a selected link" which was done already years before. The novelty of using part of using landslide susceptibility maps needs a lot of improvement.

1. Are these up to international standards?

   See above.

2. Are the scientific methods and assumptions valid and outlined clearly?

   Partially. The used model for the creation of the landslide susceptibility map and the agent-based traffic model is explained well, but the interaction of both of them is not.

3. Are the results sufficient to support the interpretations and the conclusions?

   No. What was done was a simple traffic simulation, which does not indicate the vulnerability of the road network to landslides. This link is missing in the current version.

4. Does the author reach substantial conclusions?

   No. at the beginning of the paper an "... agent-based vulnerability assessment of rural road networks ..." was promised, but it ended up in a simple traffic simulation. No conclusions about the actual vulnerability were made.

5. Is the description of the data used, the methods used, the experiments and calculations made, and the results obtained sufficiently complete and accurate to allow their reproduction by fellow scientists (traceability of results)?

   Yes. The authors put emphasis on open source data and open source software. However, I would like to encourage the authors also to publish their code, so that fellow scientists actually can reproduce their results.

6. Does the title clearly and unambiguously reflect the contents of the paper?

   No. The title indicates a vulnerability assessment, which was not done.

7. Does the abstract provide a concise, complete and unambiguous summary of the work done and the results obtained?

   No. The abstract indicates topics which are not treated at all in the paper e.g. "The focus of this case study is on resilience issues and support for decision making in the context of a large-scale sectoral approach."

8. Are the title and the abstract pertinent, and easy to understand to a wide and diversified audience?

   See above, both have to be adapted!

9. Are mathematical formulae, symbols, abbreviations and units correctly defined and used? If the formulae, symbols or abbreviations are numerous, are there tables or appendixes listing them?

   No mathematical formulae, symbols, abbreviations were used in this paper.

10. Is the size, quality and readability of each figure adequate to the type and quantity of data presented?

    Partially. The content of the figures can be enhanced so that the reader gets more information about the investigated area (see point 4).

11. Does the author give proper credit to previous and/or related work, and does he/she indicate clearly his/her own contribution?

    Yes.

12. Are the number and quality of the references appropriate?

    Yes. The authors have done an extended literature review.
13. Are the references accessible by fellow scientists?

    Yes.

14. Is the overall presentation well structured, clear and easy to understand by a wide and general audience?

    The structure of the paper is okay, but the content has to be improved.

15. Is the length of the paper adequate, too long or too short?

For reading the length of the paper is okay, however, it could be shortened (see below).

1. Is there any part of the paper (title, abstract, main text, formulae, symbols, figures and their captions, tables, list of references, appendixes) that needs to be clarified, reduced, added, combined, or eliminated?

   As indicated in comment 5, the authors stated that "The official hazard map already provides a reasonably accurate and consistent basis for the purpose of identifying vulnerable sections." This raises the question why a new set of susceptibility maps had been created? Using the existing ones would reduce 4 pages, which can be used to describe in more detail how a researcher can come from a susceptibility map to a damaged road intersection.

   As mentioned in comment 7, the use of a 10% traffic scenario is not necessary at all and can be removed from the paper, since the results are given for the more accurate 30% traffic scenario anyway.

2. Is the technical language precise and understandable by fellow scientists?

   To avoid misunderstanding the authors should state their (used) definition of vulnerability at the beginning of the paper. Also, they should use a consistent (and approved) terminology for risk, resilience, vulnerability, consequences, . . .

3. Is the English language of good quality, fluent, simple and easy to read and understand by a wide and diversified audience?

n/a

4. Is the amount and quality of supplementary material (if any) appropriate?

n/a

———————————————

---

## Author Comment (AC1) · 14 Jun 2018

We would like to thank the referee for the thorough evaluation of our manuscript and the provided feedback. All of the feedback provided will certainly contribute to improve the quality of the manuscript.

Please find our responses below, with referee comments in italics, and authors' responses in standard format.

[Figure]

**1 General comments**

1. *In the tile, the paper implies an "... agent-based vulnerability assessment of rural road networks...". Unfortunately, this is not delivered, i.e. at the end of the paper, the reader does not have a clear answer on the question "How vulnerable is the road network (in Vorarlberg) towards landslide hazards?" Instead, 10 (very specific) Scenarios were analyzed and compared, considering the detour length and the evasion time. Hence, the concept how to assess network vulnerability has to be better expressed, beginning with a clear definition how the authors define vulnerability in their paper, and the introduction of some vulnerability measures for quantification and justification (e.g. using indices, curves, tables, maps, etc.).*

   The reviewer is right. We will adjust the title of the manuscript to indicate that the focus of this work is on the application of an agent-based transport model rather than on a vulnerability assessment of the network. The goal of this manuscript is actually to assess the vulnerability of the **agents**, not the vulnerability of the road network. Thus, the purpose of this manuscript is to demonstrate this approach, thereby highlighting the benefits of obtaining in-depth conclusions using spatio-temporally disaggregated mobility behaviour. See response to (3) for a more detailed description on that aspect. We will indicate this more clearly in the manuscript, including a definition of vulnerability as used in this context. In addition, we will adjust the introduction/motivation accordingly.

2. *Throughout the introduction, several times the importance of socio-economic impacts and the severe losses caused by the disrupting services are mentioned. Finally, only the prolongation of the travel is considered. Since the authors already implemented a very detailed agent-based model with many socio-demographic data (e.g. the agents are employed/unemployed), why not actually assessing the socio-economic impacts? This would also be a novel contribution of the paper, which is currently missing.*

The reviewer is right that this is an important issue in this context. We will add a more detailed analysis of select socio-demographic data (e.g. age, employment status) in the results section. Furthermore, we will re-phrase our introduction in terms of including a broader discussion on the use of ABM in natural hazards management. However, we would like to point out that although the underlying mobility patterns of the modelled agents are available, the transfer from their individual socio-economic features to comprehensive socio-economic impact values and resulting costs entails an extensive analysis in itself, which is methodologically different from the interdisciplinary basis that was presented in this paper. The travel time costs nevertheless provide a quantifiable indication that can serve as the basis for further analyses. The novel contribution of this work was to demonstrate the applicability of using spatio-temporally disaggregated mobility behaviour data for assessing the impacts on the local population rather than providing monetary quantification of impacts. We will discuss that in the manuscript and adjust the outlook section accordingly. In addition, we will emphasize the benefits and novelty of the approach by including daily evasion time heatmaps, illustrating the spatio-temporally disaggregated conclusions that only an agent-based approach allows for.

3. *In the current version of the paper, the methodology can be summarized as: "Running a traffic model, thereby disabling a selected link", which is not a novel concept, or method. What's missing is a clear description of the novelty for the proposed methodology. i.e. What is new? How is it better than other approaches?*

As mentioned in (2), we will highlight the novelty and the advantages of using an agent-based traffic model instead of a conventional (macroscopic) traffic model or even a simple network analysis. While several studies on network effects caused by disabled links do exist, we are not aware of a similar analysis that has been performed using an agent-based traffic model. The drawbacks of classical

aggregated (macroscopic) traffic models include:

- everything is just an indistinguishable flow → "gravity model ambiguities"
- modal re-decision can not be mapped w.r.t. each agent's properties
- flows are usually zone-to-zone, as larger spatial aggregation areas are used
- models are just trip-based, considering single "hops" from activity to activity, not whole day-plans
- more time-averaging and space averaging, therefor less policy-sensitive

As discussed above, the approach described in this manuscript alleviates such problems by employing an agent-based traffic simulation, which works on a disaggregated (finer) scale. We will clarify and emphasize this aspect in our method section in the manuscript.

4. *Concerning the landslide susceptibility map:*

   a *There is a conceptual flaw, using landslide susceptibility maps for assessing network-related processes. Contrary to building assets (e.g. houses, facilities, etc.), networks are used to describe dynamic processes (e.g. traffic flow), with the consequence that local events can have a severe impact on the whole network (as the authors showed in their example). The problem is that landslide susceptibility maps describe only the relative likelihood of future landslides, however, since the network is more than its components, there is a probability that a very unlikely landslide (low susceptibility), causes more harm than a very likely landslide. For example, there is a very low susceptibility that a landslide will be triggered and affecting a major connection, causing thousands of people to stay at home while incident 6 affects only 128 agents. At the current state of the paper, such network effects are completely neglected, however, this is the core concept (and challenge) of analysing network structures.*

The selected incidents are used as scenarios to illustrate the applicability of the traffic model. To assert the realism of our assumptions, blockage locations were chosen based on the landslide susceptibility map. The goal was not to assess the worst effects (i.e. focusing on extreme landslides events), but the most likely blockages representing realistic everyday risks in this area (often a result of highly variable but very local thunderstorms during summertime), which in consequence were used as basis for assessing the impact on agents. We will clarify this in the manuscript.

b *Why did the authors develop a landslide susceptibility map although "The government of the province of Vorarlberg offers an official landslide susceptibility map..." (Page 5/ Line 26) and "The official hazard map already provides a reasonably accurate and consistent basis for the purpose of identifying vulnerable sections." (11/30)? Additionally, the used Weight of Evidence Method (Bonham-Carter, 1994) is nothing new, and therefore worth to spent 4 pages of the paper, only to figure out that the official landslide susceptibility map is a good enough estimate.*

Against the background of publication bias, we feel it to be our responsibility as scientists not to hide negative results, but rather to discuss them openly. In this case, our assumption that the application of the Weight of Evidence method would clearly improve the susceptibility map (as implied by various publications on this well-known method), was not met. Instead, our efforts to improve the official susceptibility map yielded only slight improvements. We found this outcome (which is largely due to the data quality of observed landslide events) to be worth mentioning and discussing. However, we do offer to improve on the landslide inventory data by performing a complete landslide mapping (polygons) from satellite images for the whole study area. Subsequently, a new landslide susceptibility map will be calculated again, using these new input data.

5. *Concerning the selection process of links to be blocked*

  a *It is absolutely not clear, how the 12 incident sides were selected. Please, give a detailed description how this was done (quantitative?, qualitative?), especially since the authors remove later on selected sides ("... 12 had to be removed due to its close proximity to Silvretta-Hochalpenstraße ..."(13/1).)*

  We agree with the reviewer that this is unclear. We will clarify the incident site selection procedure.

  b *Also, it would be of interest which landslide susceptibility is associated with each incident side. See point 5: If only areas with high susceptibility are considered, the question arises, if there is not a scenario where the road network is more vulnerable to landslides in less susceptible areas.*

  We will include the landslide susceptibility values for each incident into Table 2.

  c *An important part missing is the interaction between landslides and road network. In the current version, it is assumed that a landslide occurs at the incident side and completely damage (block) the road section over a period of at least 24 hours (runtime for the MATSim model)? If so, these are very strict assumptions and is contradicted by the author's statement "...due to the fact that landslides, which affected traffic routes or (agri-)cultural areas, are usually fixed quickly and efficiently." (12/10). Also, how could such assumptions be made without the knowledge of the particular landslide type (initiation and run-out, volume, speed)? The likelihood of the occurrence of landslides is not a sufficient reason to assume a damaged infrastructure. Please, specify the assumptions made and give a detailed description how the (physical) damage of the infrastructure was derived from a landslide susceptibility map.*

  The reviewer raises an interesting point in pointing out that the likelihood

of the occurrence of landslides is not a sufficient reason to assume a damaged infrastructure - especially, if in-depth knowledge of the event is lacking. However, we would like to emphasize again that the focus of this study is on the application of an agent-based traffic model to model responses in case of capacity reductions of a regional scale road network. We argue that occurrence probability of landslides is a reasonable proxy for assuming likely network interruptions that are representative of common, everyday risks in the study area. While the primary road network is indeed very resilient to landslide exposure, rural road networks are way more susceptible to landslide occurrences (since the high building standards for highways cannot be met on all rural roads). Complete interruptions caused by landslides are just one possible scenario to obtain blockage points. The described methodology also allows to specify capacity reductions (e.g. 50% capacity if only one lane is blocked). We would like to point out that we put the focus on assessing the whole federal state using several likely incident locations with a predefined (simple) interruption scenario instead of focusing on one or two locations with different varying blockage duration and capacity reduction patterns. Please note that the statement that roads are "usually fixed quickly and efficiently" does not mean that roads are fixed instantly. For safety reasons (and of course the FRC of the road), interruptions of at least 1-2 days are common for high level roads. Interruptions of several days are common on the rural road network in Austria. This is well within the assumptions made in this study. Finally, modelling physical damage is no focal topic of this manuscript. Landslides are merely considered as a scenario to obtain blockage points.

6. *Concerning the agent-based traffic model*

   a *The implementation of an agent-based model is very ambiguous, please clearly state why such an approach was used, especially since most of the*

*results (affected persons, detour lengths, evasion times) could also be observed by a flow-based traffic assignment.*

We can understand the confusion regarding the differences, but have to underline that the comment of the reviewer is not true as stated. We will clarify this as requested by the reviewer. The main benefit is in the temporal and spatial disaggregation of information on agents, which are lost in the flow of conventional transport models. See response to (3).

b *In the current version of the paper, several assumptions made and several limitations of the traffic model are not clearly stated. e.g. the MATSim simulation considers only/maximum one day, an agent has perfect knowledge of the interrupted section, origin and destination do not change during and after extreme events, etc.*

This is only partly correct. The agents do not have perfect knowledge of the interrupted section initially. They only acquire this knowledge iteratively as a whole population after optimizing for best route user-equilibria. Most of the limitations are discussed in the manuscript (18/6ff). We will further clarify these aspects mentioned by the reviewer.

c *A major shortcoming is that only trips of inhabitants of Vorarlberg are considered, which does not reflect reality and certainly leads to an underestimation of the socio-economic impacts in the region. The question is how can the vulnerability assess given this constraint? Additionally, how could the traffic model be calibrated and validated, neglecting a majority of the travellers on the network?*

This may indeed be considered a shortcoming. However, it is also clearly stated as such in the manuscript. Conducting mobility surveys is extremely resource-intensive. In the said case study region, we would need similar mobility survey data for Germany (Bavaria), Switzerland (Cantons of St. Gallen and Grisons), Liechtenstein and the Austrian federal state of Tyrol to cover

all adjacent countries/regions. Given the resources at hand, and since the focus is on the impacts of network interruptions on the local population (user level), we consider this model to be useful despite certain restrictions (i.e. likely underestimation of the impacts). Also, the fidelity of any model is restricted toward the boundary areas. In addition, it can hardly be argued that "a majority of travellers on the network" are neglected. Only the two highways, A14 and S16, can be considered major transit routes in the area, a majority of commuter travel on rural roads is definitely captured by the underlying mobility survey data.

d *Why was so much focus put on introducing and analyzing 10% scenarios, without any additional benefit for doing so? It could have been stated, that for computational reasons a pre-sampling with 10% of the agents has been done, but the evaluation has been done with a 30% scenario.*

The official guidelines state that 10% is a reasonable subset of the full population to model all relevant effects. However, results of the subsequently used 30% sample show different implications, as discussed in the manuscript. This is particularly the case for the variance of the results, which increases (!) with increasing sample size. We consider this to be an interesting discovery, since this questions the general recommendations. We will clarify this in the manuscript.

e *It is not clear how many simulations (not iterations) have been done for each incident. In other words, how often was the traffic model run for one incident? Since the agent-based model tries to optimize the behaviour of multiple agents, the simulation results might change over time.*

For computational reasons, each simulation was only done once. The model could be re-run multiple times using different random seeds. While this might provide better insights with respect to analyses of specific incidents by reducing uncertainty of the results, this does not affect the applicability of the

demonstrated approach. We will add this information to the manuscript.

f *Using advanced modelling tools often suggests precise outcomes, however, since many unknown input parameters are necessary, the results might come with high uncertainties. These uncertainties have to be quantified in order to make meaningful statements. At least a more detailed (quantitative) description how accurate the traffic model compared to the actually measured traffic volumes should be given.*

The core purpose of any traffic model tool is to provide predictive models (based on partially known real-world data) for scenario estimation, rather than precisely calculate exact values. Reliability based on comparisons with traffic count data is also limited, since these data are only valid for a very specific location and are likely to change at the next crossroads. In addition, traffic count data are also subject to high uncertainties (as they are often extrapolated from a measurement period of e.g. 2 weeks). Also, KPI values for assessing the quality of traffic models are not yet available, but currently still under development. The whole question can thus be broken down to "systematics vs. statistical uncertainties". We argue that discussing systematics is more important in this context. The uncertainties of any traffic model are rather grounded in the quality of the input data rather than in the model itself. Therefore, classical uncertainty quantification (e.g. in terms of Monte Carlo simulation using multiple model runs to obtain a distribution of results) often does not provide substantial insights. In addition, this kind of uncertainty assessment would be computationally prohibitive for the present study when using a sufficiently large number of scenario runs for all incident scenarios.

However, we consider this issue a valuable question which is intended to be answered in future work.

7. *As mentioned in the beginning, it is hard to interpret the results and conclude*

*how vulnerable the road infrastructure is. For example for side incident 10, 4709 agents are affected by an average evasion time of 3:10 minutes over a whole day. Does this mean there is almost no vulnerability against landslides? How can road authorities derive conclusions from this results? Should they invest in some protection measures or not?*

The first order interpretation of this result is correct. Road blockage of incident 10 has only minor effects on the traffic displacement in this area. We will add an additional subsection on vulnerability assessment to the discussion section to explore this issue further in the manuscript.

**2 Specific comments/questions**

1. *(1/10): "The focus of this case study is on resilience issues and support for decision making in the context of a large-scale sectoral approach." this is clearly not the case in this paper. Either this will be added to the paper of this statement should be deleted.* The reviewer is right. We will adjust this statement accordingly.

2. *(1/15) Here only single events are considered, however, in reality, we often have to deal with the occurrence multiple hazard events (e.g. heavy rainfall caused several landslides). How can the proposed methodology cope with such situations?*

   From a methodological point of view, this is not different from what was done in the present study. Expanding the methodology accordingly is simple: instead of removing a single link from the routing graph, multiple links can be removed, and the model can be re-run on these modified graphs.

3. *(7/23) "Road capacity was derived from the functional road class." For personal interest, how was this done and in which range where those values per road*

*class?*

Road capacity was estimated from the FRC attribute of the OSM graph:

```
def get_capacity_per_lane(frc):
    if frc == 0:
        return 2000
    if frc == 1:
        return 1500
    if frc == 2:
        return 1200
    if frc == 3:
        return 1000
    else:
        return 500
```

4. *(10/1) Figure 1. The different road classes should be indicated (e.g. highway, primary road, ...) in order to give the reader an overview how the network is structured and where the major links are located. Additionally, since the base scenario is already computed, a map with the traffic volume should be added, to indicate the traffic flow. Next, to such a figure, it would be interesting to see a figure for the traffic volumes of an interrupted network.*

   We will add information on road classes as well as maps as proposed by the reviewer. In addition, we intend to provide additional supplementary material in an openly accessible repository, including e.g. time-lapse videos of traffic flow across the whole day.

5. *(14/3) "In some situations, the blockage of a non-redundant link can occur, meaning that no alternative routes are available, as is the case for incident 11. Here, it is of no benefit to run a traffic simulation on the modified road graph affected by the landslide event." Actually, what would happen is that the overall travel time*

*will decrease for the network since fewer people are on the roads. The issue of missed trips (people who are cut off from the network) is neglected in the current version of the paper, however, it is important problem and should also be treated. Especially since this could cause more socio-economic impacts than a trip prolongation of several minutes.*

While this is theoretically correct we do argue that this is only of minor relevance. For a vast majority of links in the network alternative routes do exist, and the rare cases where complete blockages would result in a complete cut-off the agent-based traffic model does not offer any additional benefit over simpler traffic models. The assessment scheme would be different than on the other scenarios.

6. *(15/9) How many agents were simulated? 30% of 260000 is 78000 and not 5518. Probably this sentence has to be clarified.*

   We will add this information to the manuscript and clarify this sentence as proposed by the reviewer.

7. *(16/1) Figure 2. Why showing the 10% and the 30% example, is there any additional value in showing and discussing the 10% example?*

   See response to (6d).

8. *(19/4) "In this paper, we have shown that agent-based traffic modelling allows gaining interesting insights into the impacts of road network interruptions on the mobility behaviour of affected communities by modelling their responses to network disturbances."This might be true but is only slightly related to the topic of road network vulnerability which was promised in the title of the paper.*

   This is correct. As mentioned above we will adjust the title of the manuscript to clarify.

Again, we would like to thank the reviewer for the thorough review and the helpful

feedback provided. These comments will certainly contribute to improve the quality of the manuscript.

Overall, we feel that meeting all suggestions by the reviewer would be beyond the scope of a single manuscript. Covering the whole methodological chain from detailed landslide process simulation, agent-based traffic modelling (including various combinations of single link and multiple link failures for different values of section vulnerability), network vulnerability assessment, socio-economic analysis of consequences and provision of decision support as well as recommendations for road authorities would simply break the mould. While we do intend to incorporate a vast majority of the reviewer's suggestions, we hope that our responses do clarify why certain suggestions cannot be met.

To summarize: Our approach is based on certain assumptions and scenarios, which allow to illustrate the application of an agent-based traffic model to obtain the consequences of network interruptions (in terms of detour statistics) on the local population, by using actual mobility survey data. This manuscript is intended to serve as a methodological blueprint covering an interdisciplinary process chain from landslide susceptibility modelling via agent-based traffic modelling to an agent-specific vulnerability assessment. Thanks to the insightful reviewer comments we will add a more concise description of the process flow, including a more detailed assessment of a selection of socio-demographic variables to illustrate the advantages of an agent-based model. We are fully aware of the fact that the approach can of course be extended, e.g. by including cross-border traffic, assessing capacity reductions instead of complete blockages or assessing multiple link failures at the same time. However, all these aspects are not in the focus of this study, as they are merely methodological extensions of the approach we present here.

---

## Referee Comment (RC2) · E. Schmaltz (Referee) · 20 Jun 2018

Review of

**On the nexus between landslide susceptibility and transport infrastructure – agent-based vulnerability assessment of rural road networks in the Eastern European Alps**

by Matthias SCHLÖGL, Michael AVIAN, Gerald RICHTER, Thomas THALER, Gerhard HEISS, Sven FUCHS and Gernot LENZ

Dear Authors,
Dear Editor,

I have reviewed the aforementioned manuscript (nhess-2018-93). The paper presents an agent-based vulnerability assessment of rural road networks in the Federal State of Vorarlberg, Austria. Therefore, approaches for the generation of landslide susceptibility are combined with transport network analysis.

In my opinion, the authors provided a very interesting study, with a largely novel approach to assess the impact of landslides on transport infrastructure in Vorarlberg. However, the paper owes in some parts of methodological details and awakes some ambiguities to the reader.

Based on the aspects I found when evaluating the manuscript, I suggest to consider the manuscript for publication after revision by the authors.

With kind regards

Elmar M. SCHMALTZ

**Scope**

The topic of the study is within the scope of the journal '*Natural Hazards and Earth System Sciences*'.

**General structure**

The authors structured the manuscript very well. I believe the study area should be explained more in detail, either in the Introduction or in the Methods.

**Content**

**Title**

The title summarises the content of the research in a good way.

**Abstract**

In my opinion, the introductory part of the abstract (P1 L1-5) is too long and could be shortened to a concise sentence that directs to the research gap and the aim of the paper (P1 L6-8).
Furthermore, I believe that the results should be presented already in the abstract in a more quantitative and discussable way (P1 L17-19), leading to a closing sentence that states the key findings of the paper.

P1 L8: '*[...] landslide events*', not '*[...] landlide_ events*'.

P1 L12: '*derivates*' is the verb. '*Derivatives*' would be the noun.

**Introduction**

The introduction embeds the research into a very broad methodological and ethical context about impacts of hazards on transport systems. I do not disagree with this, however, I suggest that the authors sharpen their scientific purposes on landslide hazards and do not divagate too much into rather remotely related hazard fields (hurricanes, terrorist attacks). A connection to these fields, e.g. as application of the presented techniques and methods on those different hazards, could be given in the outlook of the study. I believe the introduction would benefit from the following structure:

(1) Introduction to transport network systems and transport network vulnerability

(2) Introduction to all kind of landslide hazards that can affect transport networks and how they can affects them in terms of topological and system-based vulnerability

(3) Introduction to the situation in Austria with focus on Vorarlberg (why was particularly Vorarlberg chosen as a study site?)

(4) Statement of the research gap, the hypothesis and related (methodological) research questions

It is up to the authors, where to present the geomorphic and infrastructural peculiarities of Vorarlberg (either in the Introduction or the Methods part). Although this paper can be considered as a methodological one, at its present form it lacks of information about the specific situations in Vorarlberg, regarding landslide dynamics and transport networks.

P2 L2: If '*Transport networks_*' is plural, then '*its*' should be '*their*'.

P2 L7: The authors mention '*a growing amount of studies*' that deal with the impact of natural hazards on roads, however, only three studies are referenced, albeit there are certainly many more. I would suggest to provide more references, at least for landslide studies that underline the purposes of this paper.

P2 L11: From a geomorphological point of view, a '*complex landscape*' does not necessarily have to be steep – just a minor comment...

P2 L14-15: What are '*reliable networks*' in this context? In general, this sentence is relatively hard to understand from my point of view.

P2 L14-20: This can be shortened to a single, concise sentence.

P2 L21-30: The aim of the paper is '[...] *to present how road infrastructure is vulnerable towards landslides [...]*'. In this paragraph, however, the authors somehow begin to embed their research into prior assessments of transport network systems that were affected either by terrorist attacks or supra-regional or national effective natural hazards like hurricanes. Even though I see the slight connection here, I am strongly suggesting to focus on what was already proposed in the abstract, which is an assessment of the impact of landslide on transport networks in Vorarlberg.

P3 L13-15: Which means it is related to (1) topological vulnerability analysis? If yes, it should be clarified explicitly.

**Data and methods**

The first subsection of section 2 (2.1 Modeling landslide susceptibility) should be restructured in a way that it follows a more logical order. The description of landslide inventories and the necessities of their compilation should be explained at first. The computation of susceptibility maps that emanate from the inventories, including the incorporation of DTM-derivatives as predictor variables within the modelling procedure, should then subsequently follow. Generally, some paragraphs appear to belong rather in the introduction part than in the methods part (e.g. P5 L4 - P7 L16). The description of the derivatives may be read like a textbook. I suggest to specifically state why these derivatives were chosen as predictors to generate the susceptibility map, with a clear focus on their geomorphic reasonability for landslide initiation. Furthermore, please explain in detail which methods were applied to compute the landslide susceptibility and provide a short description of these methods. If solely the 'Gefahrenhinweiskarte' of the Federal State Vorarlberg was used, then the authors should explicitly state that in the methods part, otherwise it is not clear to the reader if a susceptibility map was generated or an existing one was used.

P5 L2: '*of of*'

P5 L26: Since the authors refer here to regional landslide inventories and landslide susceptibility analysis, I suggest to replace '*Schmaltz et al., 2016*' with

Schmaltz, E. M., Steger, S., Glade, T. The influence of forest cover on landslide occurrence explored with spatio-temporal information, Geomorphology, 290, 250-264, https://doi.org/10.1016/j.geomorph.2017.04.024, 2017.

since a more complete landslide inventory was used.

P6 L8: It is mentioned that the landslide inventory differentiates several process groups. Which are they? Are all kinds of landslides considered (soil creep, debris flows, rockfalls) or only those of the slide-type movement? The landslide process, which is considered in the inventory should be specified in order to understand the susceptibility map.

P6 L9: '*1178 landslide were available*': Are they equally distributed? Are there any (systematic) biases that the authors detected or expected within the dataset?

P6 L11-12: Please specify the classification of the different geological units. Which of the units were considered as similar according to their lithological and geotechnical characteristics? Did the authors also distinguish between the landslide process that can be induced by different lithologies in Vorarlberg (e.g. rather steep walls in sand- or limestones in the Montafon, Rätikon, Walgau and Großwalsertal (etc.), prone to rockfalls; claystones, marls (Walgau, Bregenzer Wald, Pfänderstock) and Molasse (Doren), prone to slides; etc...)?

P6 L17: Which ALS-DTM was used? 2004? If yes, why did the authors not consider the ALS-DTM of 2011, since there were remarkable changes of both landslide dynamics (e.g. triggering event of 2005) and infrastructural development on landslide-prone hillslopes.

P6 L18: The grid sizes are confusing me. Which one was used, 5 m or 10 m? If latter, then please correct on P6 L1, or further explain why the resampling procedure was performed as mentioned in the manuscript.

**Results**

In parts, the results section reads like a mix of methods and discussion part (e.g. P9 L30 - P11 L19), I believe the authors should be much more quantitative in presenting their results and shift any interpretation into the discussion part.

P11 L5: Syntax (*'[...] and landslide areas and are [...]'*)

P11 L10-11: How did the authors deal with the detected inventory incompleteness mentioned in the manuscript?

P11 L12: Syntax (*'[...] used needs [...]'*)

P11 L12: A 50 m buffer around points that mark locations of landslide initiation introduces a large systematic error (that obviously already exists in the inventory) to the modelling procedure. The authors should justify i) why they chose such a large radius and ii) how they believe that they can still ensure geomorphic plausibility of their approach.

P11 L26: What landslide susceptibility value did the authors expect?

P11 L29-31: I believe this statement should be justified quantitatively, since no quantitative measures or values were provided by the authors that indicate a reasonable accuracy.

**Discussion**

P16 L3-5: These are two crucial points for assessing the reliability of the susceptibility maps. Although the authors identified these drawbacks, I suggest to add information on how they cope with the resulting susceptibility maps and in which way their results have to be evaluated by the reader.

P16 L6-8: Even though the geological map might be too coarse for a reliable susceptibility analysis, the authors mentioned that they were able to detect incident points along the traffic network. If geology is believed to be of central importance for landslide susceptibility*, then incident points could be detected with the rough geological map and susceptibility could be re-computed using the more detailed maps for areas where they are available.

*From my point of view, the lithological underground is a discussable predictor, since the lithological setting in Vorarlberg largely determines the topographical situation, meaning that for instance sandstone facies are responsible for steep terrains in the flysch zone, marly substrates for shallower slopes. Thus, the inclusion of slope steepness as a predictor variable might be already enough in order to avoid systematic biases in the modelling procedure. In my opinion, soil material plays a more important role and should be rather considered as predictor compared to geology. However, this is only my personal opinion that I thought be worth to mention here.

P17 L9-10: Is this always true for all rural areas throughout the year? I am thinking of locations for winter sports, which are frequent in Vorarlberg (Montafon, Bregenzerwald, etc.). Would not these areas might be also quite frequently accessed via roads and enhance an element at risk, particularly in early spring, where snowmelt occurs but winter sport tourism is still active?

**Conclusions**

P19 L8-9: The authors should provide information, which of the analysed transport systems or roads (according to their applied classification) are mostly prone to landslides. Additionally, the temporal differences at which time each type of road is mostly vulnerable would be interesting.

**Style and formatting**

**Text**

Besides some typing and syntax errors, the text is written in good English and easy to read.

**Figures and tables**

Fig. 1: A small overview map of Austria with indication where Vorarlberg is located would be helpful for readers that are not familiar with the Alps.

**NHESS specific review criteria**

1. Does the paper address relevant scientific and/or technical questions within the scope of NHESS?

   Yes.

2. Does the paper present new data and/or novel concepts, ideas, tools, methods or results?

   Yes. The analysis of transport network vulnerability towards landslides by means of agent-based modelling as it is performed in the present study depicts a novum.

3. Are these up to international standards?

   Yes.

4. Are the scientific methods and assumptions valid and outlined clearly?

   The manuscript owes a more detailed description of the landslide susceptibility analysis performed.

5. Are the results sufficient to support the interpretations and the conclusions?

   Partly, the conclusions do not necessarily reflect the findings. This is largely due to lack of quantification of the results.

6. Does the author reach substantial conclusions?

   Although it is announced in the abstract, the authors did not provide hints for decision makers based on their findings.

7. Is the description of the data used, the methods used, the experiments and calculations made, and the results obtained sufficiently complete and accurate to allow their reproduction by fellow scientists (traceability of results)?

   The methods applied partly lack on details and do not make it possible to reproduce similar results in the same way as it is presented in the manuscript.

8. Does the title clearly and unambiguously reflect the contents of the paper?

   On the whole, yes.

9. Does the abstract provide a concise, complete and unambiguous summary of the work done and the results obtained?

   The findings, interpretations and resulting conclusions should be expanded in the abstract.

10. Are the title and the abstract pertinent, and easy to understand to a wide and diversified audience?

    Yes.

11. Are mathematical formulae, symbols, abbreviations and units correctly defined and used? If the formulae, symbols or abbreviations are numerous, are there tables or appendixes listing them?

    No equations can be found in the manuscript.

12. Is the size, quality and readability of each figure adequate to the type and quantity of data presented?

    Yes. However, more concise figures would support and visualise the described results in a better way.

13. Does the author give proper credit to previous and/or related work, and does he/she indicate clearly his/her own contribution?

Yes.

14. Are the number and quality of the references appropriate?

    Yes.

15. Are the references accessible by fellow scientists?

    Yes.

16. Is the overall presentation well structured, clear and easy to understand by a wide and general audience?

    Yes.

17. Is the length of the paper adequate, too long or too short?

    It could be shortened at some parts by replacing text passages that appear to be not very comprehensive (details provided in the review).

18. Is there any part of the paper (title, abstract, main text, formulae, symbols, figures and their captions, tables, list of references, appendixes) that needs to be clarified, reduced, added, combined, or eliminated?

    Explanations are given in the detailed review.

19. Is the technical language precise and understandable by fellow scientists?

    Yes, except of minor syntax and typing errors.

20. Is the English language of good quality, fluent, simple and easy to read and understand by a wide and diversified audience?

    Yes.

21. Is the amount and quality of supplementary material (if any) appropriate?

    There is no supplementary material provided.

---

## Author Comment (AC2) · 10 Aug 2018

We would like to thank Elmar Schmaltz for the thorough evaluation of our manuscript, his feedback, and his suggestions for improvement.

Please find our responses below, with referee comments in italics, and authors' responses in standard format.

Please note that reviewer comments referring to syntax and typing errors are not answered explicitly, these will of course be corrected.

**1 Introductory comment from the authors**

To start off with we would like to state a general proposition which affects a majority of reviewer comments in this review.

The susceptibility map we used as a basis is the so-called "Gefahrenhinweiskarte Rutschungen 1:200 000 der Österreichischen Bundesländer" by Schindlmayr et al., 2016.[1] In this official data set, landslide susceptibility is derived from a very simple disposition map (based on lithology) and event data.

Therefore, the WoE approach was pursued by the authors in order to provide a more accurate, sophisticated susceptibility map. Due to the incompleteness of the underlying landslide inventory data this approach did not provide as much additional information as initially expected.

Hence, we propose to conduct a full landslide inventory for the whole federal state of Austria based on satellite images and DEM data by manually mapping the extent of previous landslides as polygons. Based on this additional information we will generate an updated susceptibility map.

**2 General structure**

*The authors structured the manuscript very well. I believe the study area should be explained more in detail, either in the Introduction or in the Methods.*

We will provide a more detailed description of the study area as suggested by the reviewer.
* * *
[1]The map can be accessed through the web-gis application eHORA (Natural Hazard Overview and Risk Assessment Austria) at http://www.hora.gv.at/. The corresponding documentation is available at http://www.hora.gv.at/assets/eHORA/pdf/2016-10-31_GHK-Rutschungen_Schlussbericht.pdf (in German).

**3 Abstract**

*In my opinion, the introductory part of the abstract (P1 L1-5) is too long and could be shortened to a concise sentence that directs to the research gap and the aim of the paper (P1 L6-8). Furthermore, I believe that the results should be presented already in the abstract in a more quantitative and discussable way (P1 L17-19), leading to a closing sentence that states the key findings of the paper.*

We will rework the abstract accordingly, including quantitative summary of the main results.

**4 Introduction**

*The introduction embeds the research into a very broad methodological and ethical context about impacts of hazards on transport systems. I do not disagree with this, however, I suggest that the authors sharpen their scientific purposes on landslide hazards and do not divagate too much into rather remotely related hazard fields (hurricanes, terrorist attacks). A connection to these fields, e.g. as application of the presented techniques and methods on those different hazards, could be given in the outlook of the study. I believe the introduction would benefit from the following structure:*

1. *Introduction to transport network systems and transport network vulnerability*

2. *Introduction to all kind of landslide hazards that can affect transport networks and how they can affects them in terms of topological and system-based vulnerability*

3. *Introduction to the situation in Austria with focus on Vorarlberg (why was particularly Vorarlberg chosen as study site?)*

[Figure]

4. *Statement of the research gap, the hypothesis and related (methodological) research questions*

*It is up to the authors, where to present the geomorphic and infrastructural peculiarities of Vorarlberg (either in the Introduction or the Methods part). Although this paper can be considered as a methodological one, at its present form it lacks of information about the specific situations in Vorarlberg, regarding landslide dynamics and transport networks.*

We would like to thank the reviewer for this constructive feedback. The introduction is rather broad indeed and does require a more precise description of the scientific purposes of the manuscript. We agree with the reviewer and will rework the introduction accordingly.

In addition, we will clarify the specific comments relating to the introduction section:

- *P2 L7: The authors mention 'a growing amount of studies' that deal with the impact of natural hazards on roads, however, only three studies are referenced, albeit there are certainly many more. I would suggest to provide more references, at least for landslide studies that underline the purposes of this paper.*
  We will add additional references as proposed by the reviewer.

- *P2 L11: From a geomorphological point of view, a 'complex landscape' does not necessarily have to be steep - just a minor comment...*
  Yes, the reviewer is right. We will adjust this accordingly.

- *P2 L14-15: What are 'reliable networks' in this context? In general, this sentence is relatively hard to understand from my point of view.*
  We agree, this sentence needs to be re-written.

- *P2 L21-30: The aim of the paper is ''[...] to present how road infrastructure is vulnerable towards landslides [...]'. In this paragraph, however, the authors somehow begin to embed their research into prior assessments of transport network systems that were affected either by terrorist attacks or supra-regional or national effective natural hazards like hurricanes. Even though I see the slight connection here, I am strongly suggesting to focus on what was already proposed in the abstract, which is an assessment of the impact of landslide on transport networks in Vorarlberg.*

  The reviewer is right, the introduction is too broad. We will focus on the research aim as proposed in the abstract.

- *P3 L13-15: Which means it is related to (1) topological vulnerability analysis? If yes, it should be clarified explicitly.*

  A topological vulnerability approach comprises the assessment of all potential impact (i.e. caused by natural hazard events) paths at the current road network system. Topological vulnerability studies are usually based on graph theoretical concepts, including behavioural aspects, such as travel demand and supply models. Here the "real" road network is represented in an albeit accurate, but still abstracted network (graph).

**5  Data and methods**

*The first subsection of section 2 (2.1 Modeling landslide susceptibility) should be restructured in a way that it follows a more logical order. The description of landslide inventories and the necessities of their compilation should be explained at first. The computation of susceptibility maps that emanate from the inventories, including the incorporation of DTM-derivatives as predictor variables within the modelling procedure, should then subsequently follow. Generally, some paragraphs appear to belong rather*

*in the introduction part than in the methods part (e.g. P5 L4 - P7 L16). The description of the derivatives may be read like a textbook. I suggest to specifically state why these derivatives were chosen as predictors to generate the susceptibility map, with a clear focus on their geomorphic reasonability for landslide initiation. Furthermore, please explain in detail which methods were applied to compute the landslide susceptibility and provide a short description of these methods. If solely the 'Gefahrenhinweiskarte' of the Federal State Vorarlberg was used, then the authors should explicitly state that in the methods part, otherwise it is not clear to the reader if a susceptibility map was generated or an existing one was used.*

We will restructure section 2.1 as suggested. However, we would like to emphasize that the description of the predictors is quite detailed on purpose, due to the potential audience of readers with non-geomorphic background. We will clarify that the susceptibility map created by the authors was used as a basis.

- *Since the authors refer here to regional landslide inventories and landslide susceptibility analysis, I suggest to replace 'Schmaltz et al., 2016' with Schmaltz, E. M., Steger, S., Glade, T. The influence of forest cover on landslide occurrence explored with spatio-temporal information, Geomorphology, 290, 250-264, https://doi.org/10.1016/j.geomorph.2017.04.024, 2017. since a more complete landslide inventory was used.*

  Thanks for pointing this out. We will replace the reference.

- *P6 L8: (i) It is mentioned that the landslide inventory differentiates several process groups. Which are they? (ii) Are all kinds of landslides considered (soil creep, debris flows, rockfalls) or only those of the slide-type movement? (iii) The landslide process, which is considered in the inventory should be specified in order to understand the susceptibility map.*

  (i) The process groups are: Mass movement (general), creep, complex large mass movement, slide, and flow. (ii) As listed in the previous answer, only slidetype landslides were considered (e.g. no falls). (iii) We will clarify this in the text
and add a short description including the number of events per process group.

- *P6 L9: '1178 landslide were available': Are they equally distributed? Are there
  any (systematic) biases that the authors detected or expected within the dataset?*
  As we point out in the result section of the manuscript (p10, L8) the mapped
  landslides are not distributed equally. We will rework the section to make this fact
  more clear.

- *P6 L11-12: Please specify the classification of the different geological units. (i)
  Which of the units were considered as similar according to their lithological and
  geotechnical characteristics? (ii) Did the authors also distinguish between the
  landslide process that can be induced by different lithologies in Vorarlberg (e.g.
  rather steep walls in sand- or limestones in the Montafon, Rätikon, Walgau and
  Großwalsertal (etc.), prone to rockfalls; claystones, marls (Walgau, Bregenzer
  Wald, Pfänderstock) and Molasse (Doren), prone to slides; etc...)?*
  (i) The concept of the Gefahrenhinweiskarte Voraralberg is as follows:

  – Lithological disposition map (scale 1:200 000) on the basis of an
    engineering-geological classification in terms of sliding susceptibility with
    three classes.
  – Event-register of landslides

  Therefore, two types of information are available:

  A: Indication if surface is prone to landslides in terms of unfavorable process
     factors (general characterization).
  B: Indication if landslides did already occur

  (ii) See comment above. We distinguished between the main different lithologies
  causing sliding events.

- *P6 L17: Which ALS-DTM was used? 2004? If yes, why did the authors not consider the ALS-DTM of 2011, since there were remarkable changes of both landslide dynamics (e.g. triggering event of 2005) and infrastructural development on landslide-prone hillslopes.*

The ALS-DTM of 2011 is used. Within the scope of the revision we suggest to use this ALS-DTM as a basis for mapping all landslides as training points on our own (like in Petschko et al., 2014, 2015).

- *P6 L18: The grid sizes are confusing me. Which one was used, 5 m or 10 m? If latter, then please correct on P6 L1, or further explain why the resampling procedure was performed as mentioned in the manuscript.*

We will clarify this in the text.

**6 Results**

- *P11 L10-11: How did the authors deal with the detected inventory incompleteness mentioned in the manuscript?*

The incompleteness of the official inventory was accepted due to practical reasons. To avoid further inconsistencies – as mentioned above – we intend to map the entire study area on our own.

- *P11 L12: A 50 m buffer around points that mark locations of landslide initiation introduces a large systematic error (that obviously already exists in the inventory) to the modelling procedure. The authors should justify i) why they chose such a large radius and ii) how they believe that they can still ensure geomorphic plausibility of their approach.*

A 50m radius was chosen to get a plausible "mean" area for modelling purposes. Slope are larger in alpine regions than in forelands, so a larger value was chosen as an assumption of slope areas in the first place. However, this will be changed by using the new training polygons obtained through mapping landslides in the whole study area.

- *P11 L26: What landslide susceptibility value did the authors expect?*

  Based on experience of our previous studies, overall occurrence probabilities are lower than expected. This is the case for both average and maximum landslide susceptibility values. For instance, much higher maximum landslide susceptibility values were expected (up to > 95% in certain cases).

- *P11 L29-31: I believe this statement should be justified quantitatively, since no quantitative measures or values were provided by the authors that indicate a reasonable accuracy.*

  Initially – due to the inconsistencies of the training points – we believed that discussing quantitative evidence is not as insightful an approach as to discuss the results qualitatively. As we were sure at the beginning that a susceptibility modeling would significantly enhance the quality of the Gefahrenhinweiskarte using the mentioned input data, we focused on the qualitative interpretation the consequences stemming from the usage of the available input data.

  As a next step – if the editor agrees – we propose to replace the WoE approach with a GAM or a tree-based classifier that will be applied to newly mapped landslides, which will feature significantly improved accuracy of landslide locations.

  We will then focus on presenting the results in a quantitative way.

**7 Discussion**

- *P16 L3-5: These are two crucial points for assessing the reliability of the suscep-tibility maps. Although the authors identified these drawbacks, I suggest to add information on how they cope with the resulting susceptibility maps and in which way their results have to be evaluated by the reader.*

  We will add a distinct description as suggested by the reviewer.

- *P16 L6-8: Even though the geological map might be too coarse for a reliable sus-ceptibility analysis, the authors mentioned that they were able to detect incident points along the traffic network. If geology is believed to be of central importance for landslide susceptibility\*, then incident points could be detected with the rough geological map and susceptibility could be re-computed using the more detailed maps for areas where they are available.*
  *\* From my point of view, the lithological underground is a discussable predictor, since the lithological setting in Vorarlberg largely determines the topographical situation, meaning that for instance sandstone facies are responsible for steep terrains in the flysch zone, marly substrates for shallower slopes. Thus, the inclu-sion of slope steepness as a predictor variable might be already enough in order to avoid systematic biases in the modelling procedure. In my opinion, soil ma-terial plays a more important role and should be rather considered as predictor compared to geology. However, this is only my personal opinion that I thought be worth to mention here.*

  We agree with the reviewer and will consider this in the revision.

- *P17 L9-10: Is this always true for all rural areas throughout the year? I am think-ing of locations for winter sports, which are frequent in Vorarlberg (Montafon, Bregenzerwald, etc.). Would not these areas might be also quite frequently ac-cessed via roads and enhance an element at risk, particularly in early spring,*

*where snowmelt occurs but winter sport tourism is still active?*

The reviewer is right. Even though this does not affect the general validity of the statement in the manuscript, we will add this aspect to the text.

**8 Conclusions**

*P19 L8-9: The authors should provide information, which of the analysed transport systems or roads (according to their applied classification) are mostly prone to landslides. Additionally, the temporal differences at which time each type of road is mostly vulnerable would be interesting.*

We will add this information as proposed by the reviewer.

**9 Figures and tables**

*Fig. 1: A small overview map of Austria with indication where Vorarlberg is located would be helpful for readers that are not familiar with the Alps.*

We will provide this map as supplement.

---

## Author Response (AR1)

**Authors' Response to Reviewer Comments**

M. Schlögl, G. Richter, M. Avian, T. Thaler,
G. Heiss, G. Lenz, S. Fuchs

November 2018

**1 Response to Reviewer 1**

We would like to thank the referee for the thorough evaluation of our manuscript and the provided feedback. All of the feedback provided will certainly contribute to improve the quality of the manuscript.

Please find our responses below, with referee comments in italics, and authors' responses in standard format.

**1.1 General comments**

1. *In the tile, the paper implies an "… agent-based vulnerability assessment of rural road networks…". Unfortunately, this is not delivered, i.e. at the end of the paper, the reader does not have a clear answer on the question "How vulnerable is the road network (in Vorarlberg) towards landslide hazards?" Instead, 10 (very specific) Scenarios were analyzed and compared, considering the detour length and the evasion time. Hence, the concept how to assess network vulnerability has to be better expressed, beginning with a clear definition how the authors define vulnerability in their paper, and the introduction of some vulnerability measures for quantification and justification (e.g. using indices, curves, tables, maps, etc.).*

   The reviewer is right. We have adjusted the title to indicate that the focus of this work is on the application of an agent-based transport model rather than on a vulnerability assessment of the network. The goal of this manuscript is actually to assess the vulnerability of the **agents**, not the vulnerability of the road network. Thus, the purpose of this manuscript is to demonstrate this approach, thereby highlighting the benefits of obtaining in-depth conclusions using spatio-temporally disaggregated mobility behaviour. See response to (3) for a more detailed description on that aspect. We have indicated this more clearly in the manuscript, including a definition of vulnerability as used in this context. In addition, the introduction/motivation section has been completely reworked.

2. *Throughout the introduction, several times the importance of socio-economic im-
pacts and the severe losses caused by the disrupting services are mentioned. Fi-
nally, only the prolongation of the travel is considered. Since the authors already
implemented a very detailed agent-based model with many socio-demographic data
(e.g. the agents are employed or unemployed), why not actually assessing the socio-
economic impacts? This would also be a novel contribution of the paper, which is
currently missing.*

The reviewer is right that this is an important issue in this context. We have
added a more detailed analysis of select socio-demographic data (e.g. age, em-
ployment status) in the results section (including 2 new plots). Furthermore, we
have reworked the introduction completely. However, we would like to point out
that although the underlying mobility patterns of the modelled agents are avail-
able, the transfer from their individual socio-economic features to comprehensive
socio-economic impact values and resulting costs entails an extensive analysis in
itself, which is methodologically different from the interdisciplinary basis that was
presented in this paper. The travel time costs nevertheless provide a quantifiable
indication that can serve as the basis for further analyses. The novel contribution
of this work was to demonstrate the applicability of using spatio-temporally disag-
gregated mobility behaviour data for assessing the impacts on the local population
rather than providing monetary quantification of impacts. We have emphasized
the benefits and novelty of the approach in the introduction, and including daily
evasion time heatmap, illustrating spatio-temporally disaggregated conclusions.

3. *In the current version of the paper, the methodology can be summarized as: "Run-
ning a traffic model, thereby disabling a selected link", which is not a novel concept,
or method. What's missing is a clear description of the novelty for the proposed
methodology. i.e. What is new? How is it better than other approaches?*

As mentioned in (2), we have highlighted the novelty and the advantages of using
an agent-based traffic model distinctively in the introduction section. While sev-
eral studies on network effects caused by disabled links do exist (see citations in
the paper), we are not aware of a similar analysis that has been performed using
an agent-based traffic model. The drawbacks of classical aggregated (macroscopic)
traffic models include:

- everything is just an indistinguishable flow → "gravity model ambiguities"

- modal re-decision can not be mapped w.r.t. each agent's properties

- flows are usually zone-to-zone, as larger spatial aggregation areas are used

- models are just trip-based, considering single "hops" from activity to activity,
not whole day-plans

- more time and space averaging, therefore less policy-sensitive

Other publications (such as e.g. Postance et al., 2017) do not consider whole-day
travel plans, but only the peak traffic flow period on the investigated network. In

addition, OD route assignment is done by a conventional network loading algorithm to find the user equilibrium. As discussed above, the approach described in this manuscript alleviates such problems by employing an agent-based traffic simulation, which works on a disaggregated (finer) scale. We have clarified and emphasized these aspects in the manuscript.

4. *Concerning the landslide susceptibility map:*

   (a) *There is a conceptual flaw, using landslide susceptibility maps for assessing network-related processes. Contrary to building assets (e.g. houses, facilities, etc.), networks are used to describe dynamic processes (e.g. traffic flow), with the consequence that local events can have a severe impact on the whole network (as the authors showed in their example). The problem is that landslide susceptibility maps describe only the relative likelihood of future landslides, however, since the network is more than its components, there is a probability that a very unlikely landslide (low susceptibility), causes more harm than a very likely landslide. For example, there is a very low susceptibility that a landslide will be triggered and affecting a major connection, causing thousands of people to stay at home while incident 6 affects only 128 agents. At the current state of the paper, such network effects are completely neglected, however, this is the core concept (and challenge) of analysing network structures.*

   The selected incidents are used as scenarios to illustrate the applicability of the traffic model. To assert the realism of our assumptions, blockage locations were chosen based on the landslide susceptibility map. The goal was not to assess the worst effects (i.e. focusing on extreme landslides events), but the most likely blockages representing realistic everyday risks in this area (often a result of highly variable but very local thunderstorms during summertime), which in consequence were used as basis for assessing the impact on agents. We have clarified this in the manuscript.

   (b) *Why did the authors develop a landslide susceptibility map although "The government of the province of Vorarlberg offers an official landslide susceptibility map..." (Page 5/ Line 26) and "The official hazard map already provides a reasonably accurate and consistent basis for the purpose of identifying vulnerable sections." (11/30)? Additionally, the used Weight of Evidence Method (Bonham-Carter, 1994) is nothing new, and therefore worth to spent 4 pages of the paper, only to figure out that the official landslide susceptibility map is a good enough estimate.*

   Against the background of publication bias, we feel it to be our responsibility as scientists not to hide negative results, but rather to discuss them openly. In this case, our assumption that the application of the Weight of Evidence method would clearly improve the susceptibility map (as implied by various publications on this well-known method), was not met in the first place. Instead, our efforts to improve the basic susceptibility map yielded

only slight improvements. We found this outcome (which is largely due to the data quality of observed landslide events) to be worth mentioning and discussing. However, our efforts during the revision to improve on the landslide inventory data by validating historic events and conducting a landslide mapping (polygons) based on the LIDAR DEM and satellite images for the whole study area did improve the quality of the map. Albeit there is still room for improvement (c.f. discussion section), the landslide susceptibility map contained in this revision is definitely more accurate than the rather simple and coarse government-provided map.

5. *Concerning the selection process of links to be blocked*

   (a) *It is absolutely not clear, how the 12 incident sides were selected. Please, give a detailed description how this was done (quantitative?, qualitative?), especially since the authors remove later on selected sides ("... 12 had to be removed due to its close proximity to Silvretta-Hochalpenstraße ..."(13/1).)*

   We agree with the reviewer that this is unclear. We have clarified this in the text (subsection 'incident sites').

   (b) *Also, it would be of interest which landslide susceptibility is associated with each incident side. See point 5: If only areas with high susceptibility are considered, the question arises, if there is not a scenario where the road network is more vulnerable to landslides in less susceptible areas.*

   We have included maximum and average landslide susceptibility values for each incident into Table 2, using 50 m buffers around all incidents.

   (c) *An important part missing is the interaction between landslides and road network. In the current version, it is assumed that a landslide occurs at the incident side and completely damage (block) the road section over a period of at least 24 hours (runtime for the MATSim model)? If so, these are very strict assumptions and is contradicted by the author's statement "...due to the fact that landslides, which affected traffic routes or (agri-)cultural areas, are usually fixed quickly and efficiently." (12/10). Also, how could such assumptions be made without the knowledge of the particular landslide type (initiation and run-out, volume, speed)? The likelihood of the occurrence of landslides is not a sufficient reason to assume a damaged infrastructure. Please, specify the assumptions made and give a detailed description how the (physical) damage of the infrastructure was derived from a landslide susceptibility map.*

   The reviewer raises an interesting point in pointing out that the likelihood of the occurrence of landslides is not a sufficient reason to assume a damaged infrastructure - especially, if in-depth knowledge of the event is lacking. However, we would like to emphasize again that the focus of this study is on the application of an agent-based traffic model to model responses in case of capacity reductions of a regional scale road network. We argue that occurrence probability of landslides is a reasonable proxy for assuming likely

network interruptions that are representative of common, everyday risks in the study area. While the primary road network is indeed very resilient to landslide exposure, rural road networks are way more susceptible to landslide occurrences (since the high building standards for highways cannot be met on all rural roads). Complete interruptions caused by landslides are just one possible scenario to obtain blockage points. The described methodology also allows to specify capacity reductions (e.g. 50% capacity if only one lane is blocked). We would like to point out that we put the focus on assessing the whole federal state using several likely incident locations with a predefined (simple) interruption scenario instead of focusing on one or two locations with different varying blockage duration and capacity reduction patterns. Please note that the statement that roads are "usually fixed quickly and efficiently" does not mean that roads are fixed instantly. For safety reasons (and of course the FRC of the road), interruptions of at least 1-2 days are common for high level roads. Interruptions of several days are common on the rural road network in Austria. This is well within the assumptions made in this study. Finally, modelling physical damage is no focal topic of this manuscript. Landslides are merely considered as a scenario to obtain blockage points. We have added a clarification to subsection 'incident sites'.

6. *Concerning the agent-based traffic model*

   (a) *The implementation of an agent-based model is very ambiguous, please clearly state why such an approach was used, especially since most of the results (affected persons, detour lengths, evasion times) could also be observed by a flow-based traffic assignment.*

   While we can understand the confusion regarding the differences, we'd like to underline that the comment of the reviewer is not true as stated. The main benefit is in the temporal and spatial disaggregation of information on agents, which are lost in the flow of conventional transport models. See responses above for futher detail. We have clarified this in the manuscript.

   (b) *In the current version of the paper, several assumptions made and several limitations of the traffic model are not clearly stated. e.g. the MATSim simulation considers only/maximum one day, an agent has perfect knowledge of the interrupted section, origin and destination do not change during and after extreme events, etc.*

   This is only partly correct. The agents do not have perfect knowledge of the interrupted section initially. They only acquire this knowledge iteratively as a whole population after optimizing for best route user-equilibria. Most of the limitations are discussed in the manuscript (18/6ff in the initial manuscript). We have further clarified these aspects mentioned by the reviewer.

   (c) *A major shortcoming is that only trips of inhabitants of Vorarlberg are considered, which does not reflect reality and certainly leads to an underestimation of the socio-economic impacts in the region. The question is how can the*

*vulnerability assess given this constraint? Additionally, how could the traffic model be calibrated and validated, neglecting a majority of the travellers on the network?*

This may indeed be considered a shortcoming. However, it is also clearly stated as such in the manuscript. Conducting mobility surveys is extremely resource-intensive. In the said case study region, we would need similar mobility survey data for Germany (Bavaria), Switzerland (Cantons of St. Gallen and Grisons), Liechtenstein and the Austrian federal state of Tyrol to cover all adjacent countries or regions respectively. Given the resources at hand, and since the focus is on the impacts of network interruptions on the local population (user level), we consider this model to be useful despite certain restrictions (i.e. likely underestimation of the impacts). Also, the fidelity of any model is restricted toward the boundary areas. In addition, it can hardly be argued that "a majority of travellers on the network" are neglected. Only the two highways, A14 and S16, can be considered major transit routes in the area, a majority of commuter travel on rural roads is definitely captured by the underlying mobility survey data.

(d) *Why was so much focus put on introducing and analyzing 10% scenarios, without any additional benefit for doing so? It could have been stated, that for computational reasons a pre-sampling with 10% of the agents has been done, but the evaluation has been done with a 30% scenario.*

The official MATSim guidelines state that 10% is a reasonable subset of the full population to model all relevant effects. However, results of the subsequently used 30% sample show different implications, as discussed in the manuscript. This is particularly the case for the variance of the results, which increases (!) with increasing sample size. We consider this to be an interesting discovery, since this questions the general recommendations. We will clarify this in the manuscript.

(e) *It is not clear how many simulations (not iterations) have been done for each incident. In other words, how often was the traffic model run for one incident? Since the agent-based model tries to optimize the behaviour of multiple agents, the simulation results might change over time.*

For computational reasons, each simulation was only done once. The model could be re-run multiple times using different random seeds. While this might provide better insights with respect to analyses of specific incidents by reducing uncertainty of the results, this does not affect the applicability of the demonstrated approach. We have added this information to the manuscript.

(f) *Using advanced modelling tools often suggests precise outcomes, however, since many unknown input parameters are necessary, the results might come with high uncertainties. These uncertainties have to be quantified in order to make meaningful statements. At least a more detailed (quantitative) description how accurate the traffic model compared to the actually measured traffic*

*volumes should be given.*

The core purpose of any traffic model tool is to provide predictive models (based on partially known real-world data) for scenario estimation, rather than precisely calculate exact values. Reliability based on comparisons with traffic count data is also limited, since these data are only valid for a very specific location and are likely to change at the next crossroads. In addition, traffic count data are also subject to high uncertainties (as they are often extrapolated from a measurement period of e.g. 2 weeks). Also, KPI values for assessing the quality of traffic models are not yet available, but currently still under development. The whole question can thus be broken down to "systematics vs. statistical uncertainties". We argue that discussing systematics is more important in this context. The uncertainties of any traffic model are rather grounded in the quality of the input data rather than in the model itself. Therefore, classical uncertainty quantification (e.g. in terms of Monte Carlo simulation using multiple model runs to obtain a distribution of results) often does not provide substantial insights. In addition, this kind of uncertainty assessment would be computationally prohibitive for the present study when using a sufficiently large number of scenario runs for all incident scenarios. However, we consider this issue a valuable question which is intended to be answered in future work.

7. *As mentioned in the beginning, it is hard to interpret the results and conclude how vulnerable the road infrastructure is. For example for side incident 10, 4709 agents are affected by an average evasion time of 3:10 minutes over a whole day. Does this mean there is almost no vulnerability against landslides? How can road authorities derive conclusions from this results? Should they invest in some protection measures or not?*

The first order interpretation of this result is correct. Road blockage of incident 10 has only minor effects on the traffic displacement in this area.

**1.2 Specific comments/questions**

1. *(1/10): "The focus of this case study is on resilience issues and support for decision making in the context of a large-scale sectoral approach." this is clearly not the case in this paper. Either this will be added to the paper of this statement should be deleted.* The reviewer is right. We have removed this statement.

2. *(1/15) Here only single events are considered, however, in reality, we often have to deal with the occurrence multiple hazard events (e.g. heavy rainfall caused several landslides). How can the proposed methodology cope with such situations?*

From a methodological point of view, this is not different to what was done in the present study. Expanding the methodology accordingly is simple: instead of removing a single link from the routing graph, multiple links can be removed, and the model can be re-run on these modified graphs.

3. *(7/23) "Road capacity was derived from the functional road class." For personal interest, how was this done and in which range where those values per road class?*

Road capacity was estimated from the FRC attribute of the OSM graph:

```python
def get_capacity_per_lane(frc):
    if frc == 0:
        return 2000
    if frc == 1:
        return 1500
    if frc == 2:
        return 1200
    if frc == 3:
        return 1000
    else:
        return 500
```

4. *(10/1) Figure 1. The different road classes should be indicated (e.g. highway, primary road, ...) in order to give the reader an overview how the network is structured and where the major links are located. Additionally, since the base scenario is already computed, a map with the traffic volume should be added, to indicate the traffic flow. Next, to such a figure, it would be interesting to see a figure for the traffic volumes of an interrupted network.*

Different classes are already represented by different line width. In addition, minor roads are omitted in the right plot. We have added two interactive leaflet maps as additional supplementary material. These include various basemaps showing different functional road classes. Two additional plots have been included, showing home locations and time delay for a selected interruption scenario.

5. *(14/3) "In some situations, the blockage of a non-redundant link can occur, meaning that no alternative routes are available, as is the case for incident 11. Here, it is of no benefit to run a traffic simulation on the modified road graph affected by the landslide event." Actually, what would happen is that the overall travel time will decrease for the network since fewer people are on the roads. The issue of missed trips (people who are cut off from the network) is neglected in the cur- rent version of the paper, however, it is important problem and should also be treated. Especially since this could cause more socio-economic impacts than a trip prolongation of several minutes.*

While this is theoretically correct we do argue that this is only of minor relevance. For a vast majority of links in the network alternative routes do exist, and the rare cases where complete blockages would result in a complete cut-off the agent-based traffic model does not offer much additional benefit over simpler traffic models. All other agents on the network might be marginally faster due to less traffic, but this is actually not the most important issue in case of total cut-off. The assessment scheme would be different than on the other scenarios.

6. *(15/9) How many agents were simulated? 30% of 260000 is 78000 and not 5518. Probably this sentence has to be clarified.*

   The number of affected agents in the baseline scenario is listed in table 1. 5518 is the total number of agents affected by any incident, corresponding to approx. 30% of $565 + 1486 + 4794 + 586 + 858 + 128 + 572 + 1404 + 576 + 4709 + 5 = 15683$, which is the total number of agents affected by any interruption in the baseline scenario. We have clarified this sentence.

7. *(16/1) Figure 2. Why showing the 10% and the 30% example, is there any additional value in showing and discussing the 10% example?*

   See response to (6d).

8. *(19/4) "In this paper, we have shown that agent-based traffic modelling allows gaining interesting insights into the impacts of road network interruptions on the mobility behaviour of affected communities by modelling their responses to network disturbances." This might be true but is only slightly related to the topic of road network vulnerability which was promised in the title of the paper.*

   This is correct. We have adjusted the title of the manuscript to avoid confusion with respect to network vulnerability versus agents' vulnerability.

Again, we would like to thank the reviewer for the thorough review and the helpful feedback provided. These comments have certainly contributed to improve the quality of the manuscript.

We have put a a lot of effort into reworking the manuscript, including a completely reworked introduction/motivation section and extensive changes to other parts of the manuscript. With some exceptions, which would simply be beyond the scope of a single manuscript, we have implemented all suggestions provided by the reviewer, which resulted in a clearer structure of the manuscript. Since covering the whole methodological chain from detailed landslide process simulation, agent-based traffic modelling (including various combinations of single link and multiple link failures for different values of section vulnerability), network vulnerability assessment, socio-economic analysis of consequences and provision of decision support as well as recommendations for road authorities would simply break the mould, we have strived to specify the aim of this paper.

Our approach is based on certain assumptions and scenarios, which allow to illustrate the application of an agent-based traffic model to obtain the consequences of network interruptions (in terms of detour statistics) on the local population, by using actual mobility survey data. This manuscript is intended to serve as a methodological blueprint covering an interdisciplinary process chain from landslide susceptibility modelling via agent-based traffic modelling to an agent-specific vulnerability assessment. Thanks to the insightful reviewer comments we will add a more concise description of the process flow, including a more detailed assessment of a selection of socio-demographic variables to illustrate the advantages of an agent-based model.

We are fully aware of the fact that the approach can of course be extended, e.g. by including cross-border traffic, assessing capacity reductions instead of complete blockages or assessing multiple link failures at the same time. However, all these aspects are not in the focus of this study, as they are merely methodological extensions of the approach we present here.

**2 Response to Reviewer 2**

We would like to thank Elmar Schmaltz for the thorough evaluation of our manuscript, his feedback, and his suggestions for improvement. Please find our responses below, with referee comments in italics, and authors' responses in standard format. Please note that reviewer comments referring to syntax and typing errors are not answered explicitly, these will of course be corrected.

**2.1 Introductory comment from the authors**

To start off with we would like to state a general proposition which affects a majority of reviewer comments in this review. The susceptibility map we used as a basis is the so-called 'Gefahrenhinweiskarte Rutschungen 1:200 000 der Österreichischen Bundesländer' by Schindlmayr et al., 2016 [1]. In this official data set, landslide susceptibility is derived from a very simple disposition map (based on lithology) and event data. Therefore, the WoE approach was pursued by the authors in order to provide a more accurate, sophisticated susceptibility map. Due to the incompleteness of the underlying landslide inventory data this approach did not provide as much additional information as initially expected. While the creation of a full landslide inventory for the whole federal state of Austria based on satellite images and DEM data would go way beyond the scope of what can be achieved within this revision, we have worked on the landslide susceptibility map by manually mapping the extent of previous landslides as reported in the historic data sets as polygons. Based on this additional information, the susceptibility map was updated accordingly.

**2.2 General structure**

*The authors structured the manuscript very well. I believe the study area should be explained more in detail, either in the Introduction or in the Methods.*

We have provided a more detailed description of the study area in the introduction section.
* * *
[1] The map can be accessed through the web-gis application eHORA (Natural Hazard Overview and Risk Assessment Austria) at `http://www.hora.gv.at/`. The corresponding documentation is available at `http://www.hora.gv.at/assets/eHORA/pdf/2016-10-31_GHK-Rutschungen_Schlussbericht.pdf` (in German).

**2.3 Abstract**

*In my opinion, the introductory part of the abstract (P1 L1-5) is too long and could be shortened to a concise sentence that directs to the research gap and the aim of the paper (P1 L6-8). Furthermore, I believe that the results should be presented already in the abstract in a more quantitative and discussable way (P1 L17-19), leading to a closing sentence that states the key findings of the paper.*

We have reworked the abstract as proposed, including quantitative summary of the main results.

**2.4 Introduction**

*The introduction embeds the research into a very broad methodological and ethical context about impacts of hazards on transport systems. I do not disagree with this, however, I suggest that the authors sharpen their scientific purposes on landslide hazards and do not divagate too much into rather remotely related hazard fields (hurricanes, terrorist attacks). A connection to these fields, e.g. as application of the presented techniques and methods on those different hazards, could be given in the outlook of the study. I believe the introduction would benefit from the following structure: 1. Introduction to transport network systems and transport network vulnerability 2. Introduction to all kind of landslide hazards that can affect transport networks and how they can affects them in terms of topological and system-based vulnerability 3. Introduction to the situation in Austria with focus on Vorarlberg (why was particularly Vorarlberg chosen as study site?) 4. Statement of the research gap, the hypothesis and related (methodological) research questions. It is up to the authors, where to present the geomorphic and infrastructural peculiarities of Vorarlberg (either in the Introduction or the Methods part). Although this paper can be considered as a methodological one, at its present form it lacks of information about the specific situations in Vorarlberg, regarding landslide dynamics and transport networks.*

We would like to thank the reviewer for this constructive feedback. The introduction was rather broad indeed and did require a more precise description of the scientific purposes of the manuscript. We agree with the reviewer and have reworked the introduction completely. In addition, we have clarify the specific comments relating to the introduction section:

- *P2 L7: The authors mention 'a growing amount of studies' that deal with the impact of natural hazards on roads, however, only three studies are referenced, albeit there are certainly many more. I would suggest to provide more references, at least for landslide studies that underline the purposes of this paper.*

  We have added additional references as proposed by the reviewer.

- *P2 L11: From a geomorphological point of view, a 'complex landscape' does not necessarily have to be steep - just a minor comment...*

  Yes, the reviewer is right. However, the term 'complex orography' or 'complex topography' is commonly used in similar studies.

- *P2 L14-15: What are 'reliable networks' in this context? In general, this sentence is relatively hard to understand from my point of view.*

  Has been clarified and re-written.

- *P2 L21-30: The aim of the paper is '[...] to present how road infrastructure is vulnerable towards landslides [...]'. In this paragraph, however, the authors some- how begin to embed their research into prior assessments of transport network systems that were affected either by terrorist attacks or supra-regional or national effective natural hazards like hurricanes. Even though I see the slight connection here, I am strongly suggesting to focus on what was already proposed in the ab- stract, which is an assessment of the impact of landslide on transport networks in Vorarlberg.*

  The reviewer is right, the introduction was too broad. It has been re-written almost completely.

- *P3 L13-15: Which means it is related to (1) topological vulnerability analysis? If yes, it should be clarified explicitly.*

  A topological vulnerability approach comprises the assessment of all potential impact (i.e. caused by natural hazard events) paths at the current road network system. Topological vulnerability studies are usually based on graph theoreti- cal concepts, including behavioural aspects, such as travel demand and supply models. Here the 'real' road network is represented in an albeit accurate, but still abstracted network (graph).

**2.5 Data and Methods**

- *The first subsection of section 2 (2.1 Modeling landslide susceptibility) should be re- structured in a way that it follows a more logical order. The description of landslide inventories and the necessities of their compilation should be explained at first. The computation of susceptibility maps that emanate from the inventories, including the in- corporation of DTM-derivatives as predictor variables within the modelling procedure, should then subsequently follow. Generally, some paragraphs appear to belong rather n the introduction part than in the methods part (e.g. P5 L4 - P7 L16). The description of the derivatives may be read like a textbook. I suggest to specifically state why these derivatives were chosen as predictors to generate the susceptibility map, with a clear focus on their geomorphic reasonability for landslide initiation. Furthermore, please explain in detail which methods were applied to compute the landslide susceptibility and provide a short description of these methods. If solely the 'Gefahrenhinweiskarte' of the Federal State Vorarlberg was used, then the authors should explicitly state that in the methods part, otherwise it is not clear to the reader if a susceptibility map was generated or an existing one was used.*

  We have restructured section 2.1, and partially shortened it as suggested. However, we would like to emphasize that the description of the predictors is quite detailed on purpose, due to the potential audience of readers with non-geomorphic background.

We will clarify that the susceptibility map created by the authors was used as a basis.

- *Since the authors refer here to regional landslide inventories and landslide susceptibility analysis, I suggest to replace 'Schmaltz et al., 2016' with*

  *Schmaltz, E. M., Steger, S., Glade, T: The influence of forest cover on landslide occurrence explored with spatio-temporal information, Geomorphology, 290, 250-264, https://doi.org/10.1016/j.geomorph.2017.04.024, 2017.*

  *since a more complete landslide inventory was used.*

  Thanks for pointing this out. We have added the reference.

- *P6 L8: (i) It is mentioned that the landslide inventory differentiates several process groups. Which are they? (ii) Are all kinds of landslides considered (soil creep, debris flows, rockfalls) or only those of the slide-type movement? (iii) The landslide process, which is considered in the inventory should be specified in or- der to understand the susceptibility map.*

  (i) The process groups are: Mass movement (general), creep, complex large mass movement, slide, and flow. (ii) As listed in the previous answer, only slide- type landslides were considered (e.g. no falls). (iii) We have clarified this in the text.

- *P6 L9: '1178 landslide were available': Are they equally distributed? Are there any (systematic) biases that the authors detected or expected within the dataset?*

  As we point out in the result section of the manuscript (p10, L8) the mapped landslides are not distributed equally. We have reworked the section to make this fact more clear.

- *P6 L11-12: Please specify the classification of the different geological units. (i) Which of the units were considered as similar according to their lithological and geotechnical characteristics? (ii) Did the authors also distinguish between the landslide process that can be induced by different lithologies in Vorarlberg (e.g. rather steep walls in sand- or limestones in the Montafon, Rätikon, Walgau and Großwalsertal (etc.), prone to rockfalls; claystones, marls (Walgau, Bregenzer Wald, Pfänderstock) and Molasse (Doren), prone to slides; etc...)?*

  (i) The concept of the Gefahrenhinweiskarte Voraralberg is as follows:

  - Lithological disposition map (scale 1:200 000) on the basis of an engineering-geological classification in terms of sliding susceptibility with three classes.
  - Event-register of landslides

  Therefore, two types of information are available:

  - Indication if surface is prone to landslides in terms of unfavorable process factors (general characterization).
  - Indication if landslides did already occur

(ii) See comment above. We distinguished between the main different lithologies causing sliding events.

- *P6 L17: Which ALS-DTM was used? 2004? If yes, why did the authors not consider the ALS-DTM of 2011, since there were remarkable changes of both landslide dynamics (e.g. triggering event of 2005) and infrastructural development on landslide-prone hillslopes.*

  The ALS-DTM of 2011 is used.

- *P6 L18: The grid sizes are confusing me. Which one was used, 5 m or 10 m? If latter, then please correct on P6 L1, or further explain why the resampling procedure was performed as mentioned in the manuscript.*

  A 10m grid size was used. We have clarified this in the text.

**2.6 Results**

- *P11 L10-11: How did the authors deal with the detected inventory incompleteness mentioned in the manuscript?* The incompleteness of the official inventory had to be accepted due to practical reasons. Efforts were undertaken to map additional landslides using satellite data and a very high resolution DEM. Conducting an extensive creation of a landslide inventory (as e.g. in Schmaltz et al., 2017) for a whole federal state would requires prohibitively large amounts of resources, even though the modelling procedure would benefit from the additional wealth of data.

- *P11 L12: A 50 m buffer around points that mark locations of landslide initiation introduces a large systematic error (that obviously already exists in the inventory) to the modelling procedure. The authors should justify i) why they chose such a large radius and ii) how they believe that they can still ensure geomorphic plausibility of their approach.*

  A 50m radius was chosen to get a plausible 'mean' area for modelling purposes. Slope are larger in alpine regions than in forelands, so a larger value was chosen as an assumption of slope areas in the first place. When mapping landslides in the whole study area, this turned out to be a quite reasonable buffer size.

- *P11 L26: What landslide susceptibility value did the authors expect?*

  Based on experience of our previous studies, overall occurrence probabilities are lower than expected. This is the case for both average and maximum landslide susceptibility values. New modelling results are more in line with our expectations.

- *P11 L29-31: I believe this statement should be justified quantitatively, since no quantitative measures or values were provided by the authors that indicate a reasonable accuracy.*

  We have included average WoE values for the affected incident links to table 2 (incident site overview) and removed this statement from the manuscript entirely.

**2.7 Discussion**

- *P16 L3-5: These are two crucial points for assessing the reliability of the susceptibility maps. Although the authors identified these drawbacks, I suggest to add information on how they cope with the resulting susceptibility maps and in which way their results have to be evaluated by the reader.*

  We have reworked the statement accordingly.

- *P16 L6-8: Even though the geological map might be too coarse for a reliable susceptibility analysis, the authors mentioned that they were able to detect incident points along the traffic network. If geology is believed to be of central importance for landslide susceptibility\*, then incident points could be detected with the rough geological map and susceptibility could be re-computed using the more detailed maps for areas where they are available.*

  *From my point of view, the lithological underground is a discussable predictor, since the lithological setting in Vorarlberg largely determines the topographical situation, meaning that for instance sandstone facies are responsible for steep terrains in the flysch zone, marly substrates for shallower slopes. Thus, the inclu- sion of slope steepness as a predictor variable might be already enough in order to avoid systematic biases in the modelling procedure. In my opinion, soil ma- terial plays a more important role and should be rather considered as predictor compared to geology. However, this is only my personal opinion that I thought be worth to mention here.*

  While the reviewer is correct that the geological underground is to some extent related to the slope, the correlation between lithology class and slope is not as high as implied by the reviewer (c.f. Figure 1).

[Figure]

Figure 1: Slope by lithology class.

Therefore, both slope and lithology class are kept in the modelling process. Since the main focus of this paper is on the impacts of interruptions on the local population, and the selection of interruption points is not solely based on the geological map, which is only one of several predictors for the susceptibility map, the added value of including two different spatial resolutions of different geological maps would be only minor.

- *P17 L9-10: Is this always true for all rural areas throughout the year? I am thinking of locations for winter sports, which are frequent in Vorarlberg (Montafon, Bregenzerwald, etc.). Would not these areas might be also quite frequently accessed via roads and enhance an element at risk, particularly in early spring, where snowmelt occurs but winter sport tourism is still active?*

  The reviewer is right. Even though this does not affect the general validity of the statement in the manuscript, we have clarified it accordingly.

**2.8 Conclusion**

*P19 L8-9: The authors should provide information, which of the analysed transport systems or roads (according to their applied classification) are mostly prone to landslides. Additionally, the temporal differences at which time each type of road is mostly vulnerable would be interesting.*

This is extremely complex and excessively prohibitive with respect to simulation efforts, as this would require to introduce in time-dependent disturbances (e.g. for each hour) and subsequent simultation of agent schedule optimization for all incidents. Daily traffic load curves are available, though.

**2.9 Figures and Tables**

*Fig. 1: A small overview map of Austria with indication where Vorarlberg is located would be helpful for readers that are not familiar with the Alps.*

We have provided this map as Fig. 1.

[revised manuscript text omitted]

---

## Author Response (AR2)

**Author's Comments to nhess-2018-93**

We would like to thank the Editor and both revievers for their thorough and constructive feedack. Please find our responses to the minor issues below:
* * *
**Editor Comments:**

Dear authors,
You provided a suitable reply to all the comments raised by the reviewers. At this point, the article should be ready for publication. I would, therefore, suggest checking two minor issues (very easy to address):
(1) Discussion and Conclusion are merged. I would recommend to separate these, leaving one short chapter (chapter. 5) to the conclusion.
(2) Figure 6: is there a legend to add showing the colour ranges?

Dear Prof. Tarolli,

we very much appreciate your positive response to the revised version of the manuscript.

ad (1):
We have separated the chapters Discussion and Conclusion as proposed.

ad (2):
We argue that a legend to Figure 6 is not really useful and of little value for the reader due to several reasons:

- These heat maps are intended to convey hot spot regions (illustrated exemplarily for two selected incidents), rather than presenting spefific data.
- Heat maps (and consequently their legends) are strongly dependend on the parametrization of the rendering algorithm (e.g. the radius used for plotting).
- Also, the units would be quite confusing in this case – for instance, for the right heat map, the unit would be *diversion time fraction of daily base travel time / ( day * person * area)"*, which is then aggregated over all agents and spatially aggregated over the heatmap raster. We have added a statement on the relative information contained in the maps ("darker means higher") to both parts of the map.

Note that for interested readers, data on home locations of affected persons of the synthetic population are provided as an interactive Leaflet map (Supplement 2).
* * *
**Report #1 (Referee #2: E. Schmaltz)**

Dear authors,
Dear Editor,

I have reviewed the revised version of the manuscript 'On the nexus between landslide susceptibility and transport infrastructure – an agent-based approach'. Based on the significant improvements that have been made compared to the first version and the high quality of the manuscript, I suggest to consider the revised version of the manuscript for publishing.

Best regards
Elmar Schmaltz

Thank you.
* * *
**Report #2 (Anonymous Referee #1)**

Dear Authors

Thank you very much for the effort you put in the revised manuscript. I think the revision has significantly improved the quality and comprehensibility of the article. I also believe that the article will be an enrichment for the community; therefore I recommend a publication in NHESS.

PS: I found two minor technical corrections which probably can be addressed during the editorial phase:

1) on page 5 line 22, please remove "(maybe citation here on one of the landslide studies)";
2) please increase the font size of Figure 3.

Thank you. We have (1) removed the passage and (2) increased the font size of figure 3.
* * *
On behalf of all co-authors,
Matthias Schlögl